# SIRT7 has a critical role in bone formation by regulating lysine acylation of SP7/Osterix

Masatoshi Fukuda [1,2], Tatsuya Yoshizawa[1], Md. Fazlul Karim[1], Shihab U. Sobuz[1], Wataru Korogi[1,2], Daiki Kobayasi[3], Hiroki Okanishi[3], Masayoshi Tasaki[4,5], Katsuhiko Ono[6], Tomohiro Sawa[6], Yoshifumi Sato[1], Mami Chirifu[7], Takeshi Masuda[8], Teruya Nakamura[7], Hironori Tanoue[9], Kazuhisa Nakashima[10], Yoshihiro Kobashigawa[11], Hiroshi Morioka[11], Eva Bober[12], Sumio Ohtsuki [8], Yuriko Yamagata [7], Yukio Ando[5,13], Yuichi Oike[9,13], Norie Araki[3], Shu Takeda[14], Hiroshi Mizuta[2] & Kazuya Yamagata[1,13]

SP7/Osterix (OSX) is a master regulatory transcription factor that activates a variety of genes during differentiation of osteoblasts. However, the influence of post-translational modifications on the regulation of its transactivation activity is largely unknown. Here, we report that sirtuins, which are NAD(+)-dependent deacylases, regulate lysine deacylation-mediated transactivation of OSX. Germline *Sirt7* knockout mice develop severe osteopenia characterized by decreased bone formation and an increase of osteoclasts. Similarly, osteoblast-specific *Sirt7* knockout mice showed attenuated bone formation. Interaction of SIRT7 with OSX leads to the activation of transactivation by OSX without altering its protein expression. Deacylation of lysine (K) 368 in the C-terminal region of OSX by SIRT7 promote its N-terminal transactivation activity. In addition, SIRT7-mediated deacylation of K368 also facilitates depropionylation of OSX by SIRT1, thereby increasing OSX transactivation activity. In conclusion, our findings suggest that SIRT7 has a critical role in bone formation by regulating acylation of OSX.

[1] Department of Medical Biochemistry, Faculty of Life Sciences, Kumamoto University, Kumamoto 860-8556, Japan. [2] Department of Orthopaedic Surgery, Faculty of Life Sciences, Kumamoto University, Kumamoto 860-8556, Japan. [3] Department of Tumor Genetics and Biology, Faculty of Life Sciences, Kumamoto University, Kumamoto 860-8556, Japan. [4] Department of Morphological and Physiological Sciences, Faculty of Life Sciences, Kumamoto University, Kumamoto 860-8556, Japan. [5] Department of Neurology, Faculty of Life Sciences, Kumamoto University, Kumamoto 860-8556, Japan. [6] Department of Microbiology, Faculty of Life Sciences, Kumamoto University, Kumamoto 860-8556, Japan. [7] Department of Structural Biology, Faculty of Life Sciences, Kumamoto University, Kumamoto 860-8556, Japan. [8] Department of Pharmaceutical Microbiology, Faculty of Life Sciences, Kumamoto University, Kumamoto 860-8556, Japan. [9] Department of Molecular Genetics, Faculty of Life Sciences, Kumamoto University, Kumamoto 860-8556, Japan. [10] Department of Pharmacology, Tsurumi University School of Dental Medicine, Yokohama 230-8501, Japan. [11] Department of Analytical and Biophysical Chemistry, Faculty of Life Sciences, Kumamoto University, Kumamoto 860-8556, Japan. [12] Department of Cardiac Development and Remodeling, Max-Planck-Institute for Heart and Lung Research, Ludwigstr. 43, 61231 Bad Nauheim, Germany. [13] Center for Metabolic Regulation of Healthy Aging, Faculty of Life Sciences, Kumamoto University, Kumamoto 860-8556, Japan. [14] Department of Physiology and Cell Biology, Tokyo Medical and Dental University, Tokyo 113-8519, Japan. These authors contributed equally: M. Fukuda, T. Yoshizawa, Md. F. Karim. Correspondence and requests for materials should be addressed to T.Y. (email: yoshizaw@kumamoto-u.ac.jp)

Bone is a multifunctional tissue with hematopoietic (stem cell niches), metabolic (mineral and energy metabolism), reproductive (male fertility), and brain (development, cognition, and behavior) functions, in addition to its basic role as a framework for the body[1–3]. Tight interplay between two types of cells, bone-forming osteoblasts and bone-resorbing osteoclasts, regulates bone remodeling, which is the process of removing older bone and replacing it with a new one. An imbalance between these cells (resorption exceeds formation) causes osteoporosis, which is characterized by impairment of bone strength that increases the risk of fracture. Osteoporosis is the most common bone disease and it is estimated that more than 200 million people suffer from it worldwide. Two transcription factors, Runt-related transcription factor 2 (RUNX2) and zinc finger transcription factor SP7/Osterix (OSX), have previously been shown to be essential for the differentiation of osteoblasts. Endochondral and intramembranous bone formation does not occur in *Runx2* knockout (KO) or *Osx* KO mice[4–6]. RUNX2 promotes skeletal development on different levels, including differentiation of mesenchymal progenitors into osteoblasts and differentiation/maturation of chondrocytes and osteoclasts. In contrast, OSX acts at a later step in the process of osteoblast differentiation, i.e., the differentiation of pre-osteoblasts into mature osteoblasts and osteocytes.

Sirtuins (SIRT1-7 in mammals) are nicotinamide adenine dinucleotide (NAD+)-dependent lysine deacylases that regulate a wide variety of biological process[7,8]. Although sirtuins were thought to only act as lysine deacylases, recent studies have revealed that these enzymes can also remove other acyllysine modifications, including propionylation, succinylation, malonylation, myristoylation, and palmitoylation. SIRT1, SIRT6, and SIRT7 are predominantly located in the nucleus, where they regulate the expression of specific genes by deacylation/deacetylation of histones and transcription factors. Previous studies have demonstrated that *Sirt1* haploinsufficient mice and two lines of osteoblast-specific *Sirt1* KO mice exhibit a reduction of bone mass that is related to decreased bone formation[9–11]. Aged mice with specific knockout of *Sirt1* in mesenchymal stem cells (MSCs) show reduction of cortical bone thickness and trabecular bone volume[12]. In addition, *Sirt6* KO mice have low-turnover osteopenia caused by impaired bone formation and bone resorption[13].

The enzymatic activity and functions of SIRT7 were poorly understood, but recent studies have revealed some important biological roles. Barber et al. reported that the acetylated K18 of histone H3 (H3 K18Ac) is a target of SIRT7, and that H3 K18Ac-specific deacetylation by SIRT7 is important for maintaining the fundamental properties of the cancer cell phenotype[14]. SIRT7 also deacetylates PAF53 to promote nucleolar transcription of ribosomal RNA[15]. Furthermore, SIRT7 acts as a histone desuccinylase with an important role in the DNA damage response and cell survival[16]. On the other hand, Yoshizawa et al. have found that *Sirt7* KO mice show resistance to induction of obesity, glucose intolerance, and fatty liver by a high-fat diet[17]. Other authors have reported on various roles of SIRT7 in the liver, heart, and adipocytes[18–20]. However, no data exist about the influence of SIRT7 on bone metabolism.

Accordingly, we here investigate the role of SIRT7 in bone metabolism by several approaches using *Sirt7* KO mice, osteoblast-specific *Sirt7* KO mice, and cell-based studies. Our findings reveal that SIRT7 is essential for bone formation by osteoblasts, and suggest that SIRT7 promotes the N-terminal transactivation activity of OSX by deacylation of lysine 368 in the C-terminal of OSX.

## Results

**Sirt7 KO mice exhibit severe osteopenia.** Deficiency of SIRT1 or SIRT6 in mice is associated with low bone mass and decreased bone formation. These findings prompted us to examine whether SIRT7 also has a role in bone metabolism, especially bone formation. To test this possibility, we first examined the bones of *Sirt7* KO mice by μCT analysis. We found reduced bone mass in female *Sirt7* KO mice aged 14–15 weeks (Fig. 1a). The cortical bone area (Ct.Ar), cortical area fraction (Ct.Ar/total cross-sectional area (Tt.Ar)), and cortical thickness (Ct.Th) were significantly lower in the femurs of *Sirt7* KO mice compared with wild-type (WT) mice (Fig. 1b–d). A significant change of mean eccentricity (Ecc), which is a parameter responsive to load-bearing, indicated that the elliptical bone shape was altered in *Sirt7* KO mice (Fig. 1e). Trabecular bone was also severely affected, with bone volume per tissue volume (BV/TV) being reduced by more than 30% in *Sirt7* KO mice compared with WT controls due to a decrease of both trabecular thickness (Tb.Th) and trabecular number (Tb.N) (Fig. 1f–h). Furthermore, the trabecular bone pattern factor (Tb.Pf, an index of inter-trabecular connectivity) was significantly increased in *Sirt7* KO mice, indicating loss of mechanical strength (Fig. 1i). Breeding of heterozygous *Sirt7* mice resulted in normal Mendelian distribution of the *Sirt7* genotype and a normal sex ratio (Fig. 1j). In addition, there were no significant differences of postnatal death, body weight, serum calcium (Ca), or serum phosphorus (P) between *Sirt7* KO mice and WT mice (Fig. 1k–m). These data indicated that the reduced bone mass of *Sirt7* KO mice was not due to premature aging or abnormalities of calcium and phosphorus metabolism. Male *Sirt7* KO mice aged 15–16 weeks showed an almost identical reduction of bone mass (Supplementary Fig. 1a–h). In the early postnatal period, 15-day old female *Sirt7* KO mice already showed similar trends of bone changes to those in adult mice (Supplementary Fig. 1i–l). Taken together, these findings indicated the development of severe osteopenia in *Sirt7* KO mice.

Next, we performed static and dynamic bone histomorphometric analysis of the lumbar spine (L4) to investigate the mechanism of osteopenia in *Sirt7* KO mice. In female *Sirt7* KO mice aged 14–15 weeks, BV/TV was reduced by nearly 30% compared with that in WT mice (Fig. 2a). The mineral apposition rate (MAR, an index of bone formation per osteoblast), bone formation rate (BFR)/TV, and mature osteoblast number (N.Ob)/tissue area (T.Ar) were all significantly reduced in *Sirt7* KO mice (Fig. 2b–d). Expression of osteoblastic marker genes was reduced in femur of *Sirt7* KO mice, including *alkaline phosphatase, liver/bone/kidney* (*Alp*), *collagen, type I, alpha 1* (*Col1a1*), *osteocalcin* (*Ocn*), *Osx*, and *Runx2* (Fig. 2e). Consistent with the mild phenotype, expression of some osteoblastic marker genes tended to be lower in *Sirt7* KO mice in the early postnatal period (Supplementary Fig. 1m). These findings demonstrated that SIRT7 regulates bone formation in vivo, suggesting that it may play a role in osteoblast proliferation and/or differentiation. In addition, the osteoclast surface (Oc.S)/bone surface (BS) and osteoclast number (N.Oc)/bone perimeter (B.Pm) were increased in *Sirt7* KO mice compared with WT mice (Supplementary Fig. 1n, o), suggesting that SIRT7 also controls osteoclastogenesis in vivo.

In aged bone, expressions of osteoblastic marker genes were extremely decreased (Supplementary Fig. 1q). Analysis of nucleic sirtuin expression revealed reduced expression of *Sirt6* and *Sirt7* in bones of aged mice (Fig. 2f). Attenuation of these sirtuins may contribute to the weakness of aged bone.

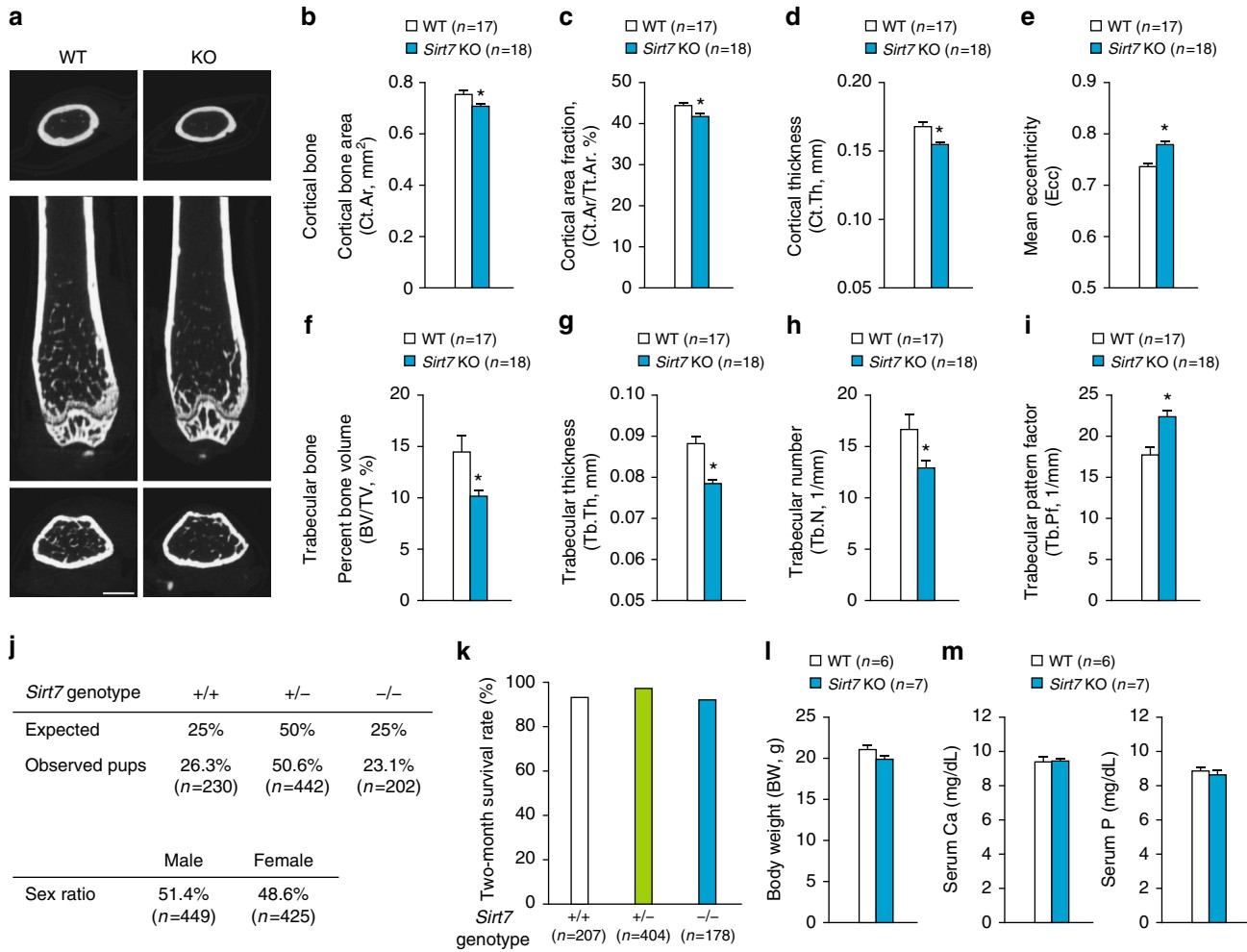

**Fig. 1** Severe osteopenia in *Sirt7* KO mice. **a–i** μCT analysis of the distal femur in female *Sirt7* KO mice and WT mice aged 14–15 weeks. Representative μCT images (**a**), Ct.Ar (**b**), Ct.Ar/Tt.Ar (**c**), Ct.Th (**d**), Ecc (**e**), trabecular BV/TV (**f**), Tb.Th (**g**), Tb.N (**h**), and Tb.Pf (**i**). **j** Mendelian frequency of the *Sirt7* genotype and sex ratio in breeding of heterozygous *Sirt7* mice. Pearson's Chi-square test indicated that the differences between WT and *Sirt7* KO mice were not statistically significant. **k** Two-month survival rates of the mice. **l**, **m** Comparison of body weight (**l**) and serum concentrations of Ca and P (**m**) between 11-week-old female *Sirt7* KO mice and WT mice. Data are shown as the mean ± SEM. Statistical significance was determined by Student's *t*-test. *$p < 0.05$ vs. WT mice (**b–i**, **l**, **m**). Scale bar, 1 mm

**SIRT7 positively regulates osteoblast differentiation**. In order to determine the intrinsic role of SIRT7 in osteoblasts, we examined the proliferation and differentiation of calvarial osteoblasts harvested from *Sirt7* KO mice and WT mice. Striking reduction of Alizarin Red S-stained mineralized nodules was observed in cultures of *Sirt7* KO osteoblasts compared with those of WT osteoblasts (Fig. 3a), although cell proliferation was not significantly different (Fig. 3b). Primary cultures of osteoblasts contain heterogeneous cells at various stages of differentiation. Next, we analyzed the effect of SIRT7 on differentiation from pre-osteoblasts to mature osteoblasts by using an osteoblastic cell line (MC3T3-E1). *Sirt7* mRNA was expressed in MC3T3-E1 cells, as well as *Sirt1* and *Sirt6* mRNAs, but *Sirt7* expression was decreased in fully differentiated MC3T3-E1 cells (Fig. 3c), suggesting a role of SIRT7 in the early stage of osteoblast differentiation. When *Sirt7* knockdown (KD) MC3T3-E1 cells were cultured for 30 days in differentiation medium, osteoblastic mineralization was markedly impaired (Fig. 3d–f), and expression of osteoblastic marker genes was significantly reduced (Fig. 3g). These results demonstrated that SIRT7 positively regulates osteoblast differentiation and thereby controls bone mineralization via a cell-autonomous mechanism.

Our bone histomorphometric analysis indicated that SIRT7 also regulates osteoclastogenesis in vivo, therefore we investigated the role of SIRT7 in osteoclastogenesis. Osteoclasts are derived from granulocyte/macrophage progenitors (GMP) in the presence of macrophage colony-stimulating factor (M-CSF) and receptor activator of nuclear factor kappa-B ligand (RANKL), which are produced by bone marrow stromal cells, osteoblasts, and osteocytes[21]. First, we performed an osteoblast-free osteoclast differentiation assay to evaluate the intrinsic role of SIRT7 in osteoclasts. Bone marrow-derived monocytes/macrophages were obtained from *Sirt7* KO and WT mice for culture with M-CSF and RANKL, after which we assessed the number of tartrate-resistant acid phosphatase (TRAP)-positive multinucleated osteoclasts. However, there were no significant differences between osteoclasts derived from monocytes/macrophages of *Sirt7* KO and WT mice with regard to numbers and expression of marker genes, such as *cathepsin K* (*Ctsk*) and *T cell, immune regulator 1, ATPase, H+ transporting, lysosomal V0 protein A3* (*Tcirg1*) (Supplementary Fig. 2a–c). Similar results were obtained by using *Sirt7* KD RAW264.7 cells, a macrophage cell line (Supplementary Fig. 2d–f). These data suggested that osteoclastic SIRT7 is not essential for in vitro differentiation of

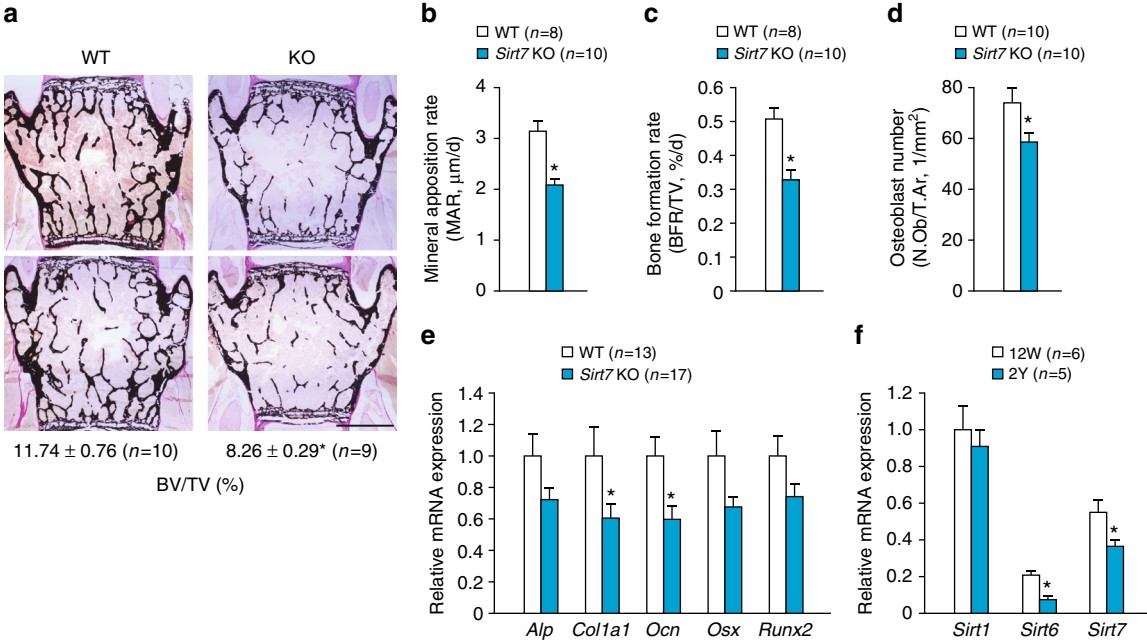

**Fig. 2** SIRT7 regulates bone formation. **a–d** Static and dynamic bone histomorphometric analyses of the lumbar spine (L4) in female *Sirt7* KO mice and WT mice aged 14–15 weeks. Representative von Kossa-stained images and BV/TV (**a**), MAR (**b**), BFR/TV (**c**), and N.Ob/T.Ar (**d**). **e** Osteoblastic marker gene expression analyzed by qRT-PCR in femurs harvested from female *Sirt7* KO mice and WT mice aged 14–15 weeks. **f** Expressions of *Sirtuins* analyzed by qRT-PCR in femurs from young (12 weeks old) and aged (2 years old) WT mice. Data are shown as the mean ± SEM. Statistical significance was determined by Student's *t*-test. *$p < 0.05$ vs. WT mice (**b–e**) or young mice (**f**). Scale bar, 1 mm

monocytes/macrophages into osteoclasts. To investigate whether SIRT7 in osteoblasts controls osteoclast differentiation, we next performed co-culture of calvarial osteoblasts isolated from *Sirt7* KO or WT mice with WT splenocytes. Osteoclast differentiation and the *Rankl/osteoprotegerin* (*Opg*) gene expression ratio were similar between *Sirt7* KO and WT osteoblasts (Supplementary Fig. 2g–j), suggesting that osteoblastic SIRT7 is not required for osteoclast differentiation in vitro.

**Osteoblastic SIRT7 is important for bone formation in vivo.**
We next generated osteoblast-specific conditional *Sirt7* KO mice (*Sirt7* osbCKO mice) to further investigate the influence of osteoblastic SIRT7 on bone formation in vivo. μCT analysis of the femur showed that bone mass was reduced in female *Sirt7* osbCKO mice aged 14–15 weeks (Fig. 4a–f). Trabecular BV/TV and Tb.Th were significantly reduced in *Sirt7* osbCKO mice compared with control mice (Fig. 4a, b), while Tb.Pf was increased in *Sirt7* osbCKO mice (Fig. 4c). In the cortical bone, Ct. Ar, Ct.Ar/Tt.Ar, and Ct.Th were all significantly reduced in *Sirt7* osbCKO mice (Fig. 4d–f). Static and dynamic bone histomorphometric analysis of the lumbar spine (L4) in *Sirt7* osbCKO female mice aged 14–15 weeks confirmed that osteoblastic SIRT7 is important for bone formation (Fig. 4g–j). On the other hand, there were no changes of Oc.S/BS and N.Oc/B.Pm (Fig. 4k, l), indicating that osteoblastic SIRT7 does not control osteoclastogenesis in vivo.

Taken together, these findings demonstrated that osteoblastic SIRT7 positively regulates the differentiation and function of osteoblasts to control ossification both in vitro and in vivo.

**SIRT7 interacts with OSX and activates transactivation.** To unravel the mechanism by which osteoblastic SIRT7 positively regulates the differentiation of osteoblasts, we investigated the effect of SIRT7 on OSX and RUNX2, which are osteoblastic master regulatory transcription factors. To assess DNA binding-

independent transcriptional activity of OSX, *Sirt7* KO and WT osteoblasts were transfected with an expression construct containing OSX fused with the GAL4 DNA-binding domain (DBD) and a luciferase reporter plasmid driven by GAL4 binding sites. In *Sirt7* KO osteoblasts, the transcriptional activity of OSX was reduced by 80% compared with that in WT osteoblasts (Fig. 5a). In contrast to OSX, there was no significant difference of RUNX2 transcriptional activity between *Sirt7* KO and WT osteoblasts (Supplementary Fig. 3a). These results were consistent with our in vitro data that expression of *Alp*, *Col1a1*, and *Osx*, which are downstream genes of OSX, was reduced in undifferentiated *Sirt7* KD MC3T3-E1 cells (Fig. 3g). In addition, attenuation of *Sirt7* expression in MC3T3-E1 cells by using siRNA led to reduced transcription of *Osx* and *Col1a1* enhancer/promoter-driven luciferase reporters, which are regulated by OSX[22,23] (Fig. 5b). These findings demonstrated that SIRT7 increases the transcriptional activity of OSX in osteoblasts in a DNA binding-independent manner. It was recently reported that OSX also acts as a transcriptional coactivator in the Dlx-directed process of osteoblast development[24]. Therefore, we assessed the effect of SIRT7 on transcription of the AT-rich motif-driven luciferase reporter, which is regulated by the Dlx5–OSX complex. As shown in Supplementary Fig. 3b, SIRT7 did not seem to have any influence on the function of OSX as a cofactor for Dlx.

Next, we examined the physical interaction between SIRT7 and OSX. When we performed a HaloTag pull-down assay using lysates of OSX-HA-overexpressing HEK293T cells, OSX strongly interacted with the Halo-SIRT7 fusion protein, but not the Halo protein (Supplementary Fig. 3c). Interaction of SIRT7 and OSX in cultured cells was also detected by the co-immunoprecipitation assay (Fig. 5c, d). Immunocytochemistry revealed that SIRT7 co-localized with OSX in MC3T3-E1 cells (Supplementary Fig. 3d). To identify the domain of SIRT7 that interacted with OSX, Halo-SIRT7 deletion mutants were incubated with lysates of HEK293T cells expressing OSX-HA and interactions were investigated by the pull-down assay. As demonstrated in Fig. 5e, the M2 region of SIRT7, which is part of

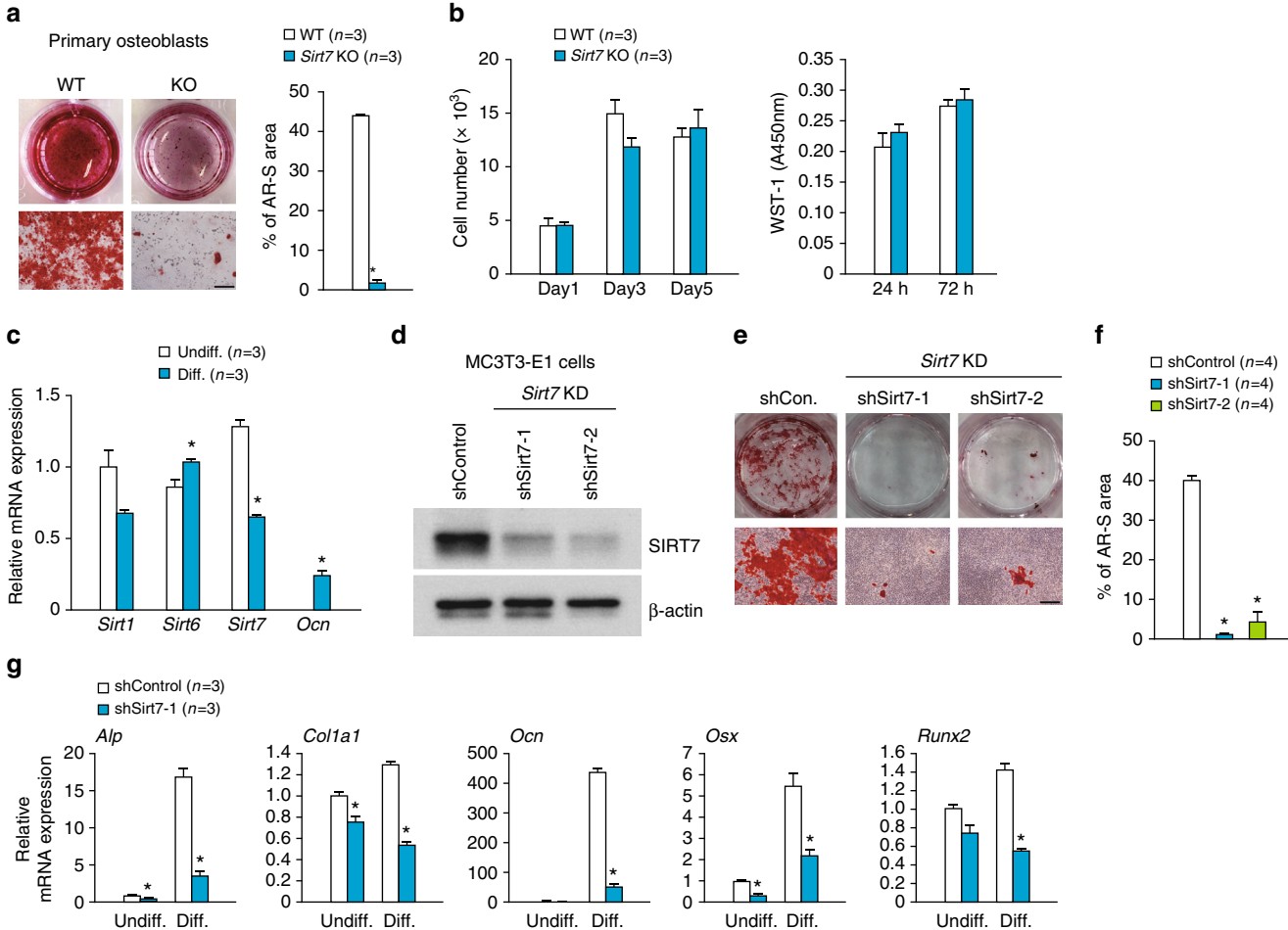

**Fig. 3** SIRT7 positively regulates osteoblast differentiation. **a** Primary calvarial osteoblasts derived from *Sirt7* KO mice and WT mice were cultured for 10 days in differentiation medium. Representative Alizarin Red S-stained images (left) and the Alizarin Red S staining area ratio (right). **b** Proliferation assay using primary calvarial osteoblasts from *Sirt7* KO mice and WT mice. Cell proliferation was measured by direct counting (left) and by the WST-1 colorimetric assay (right). **c** Expression of *Sirt1, Sirt6, Sirt7,* and *Ocn* in MC3T3-E1 cells analyzed by qRT-PCR. **d–g** MC3T3-E1 cells with *Sirt7* knockdown and control cells were cultured for 30 days in differentiation medium. Confirmation of knockdown by western blotting (**d**), representative Alizarin Red S-stained images (**e**), Alizarin Red S staining area ratio (**f**), and expression of osteoblastic marker genes analyzed by qRT-PCR (**g**). Undiff.: cells cultured in growth medium for 3 days. Diff.: cells cultured in differentiation medium for 30 days. Data are shown as the mean ± SEM. Statistical significance was determined by Student's *t*-test. *$p < 0.05$ vs. WT (**a, b**), Undiff. (**c**), or shControl (**f, g**). Scale bar, 400 μm

a conserved NAD-binding and catalytic domain known as the sirtuin core domain, showed strong binding to OSX.

Next, we performed the pull-down assay with acylated or underacylated OSX. Since binding affinity between an enzyme and its substrate generally decreases after the enzymatic reaction has finished, we considered that acylated OSX would show stronger binding to SIRT7 than deacylated OSX. As expected, endogenous OSX derived from MC3T3-E1 cells treated with nicotinamide (NAM), which inhibits the deacylase activity of sirtuins, was bound to Halo-SIRT7 beads, but endogenous OSX derived from MC3T3-E1 cells without NAM treatment showed little binding (Supplementary Fig. 3c). These results suggested that an enzyme–substrate relationship exists between SIRT7 and OSX. We also found an interaction between Halo-SIRT7 beads and OSX in HEK293T cells without NAM treatment, presumably due to insufficient deacylation of OSX by its overexpression (Supplementary Fig. 3c). Furthermore, we studied the effect of NAM on transcriptional activity of OSX. When an osteosarcoma cell line (U2OS), a preosteoblastic cell line (MC3T3-E1), and an embryonic mesenchymal stem cell line (C3H/10T1/2) were exposed to NAM for 24 h, GAL4DBD-OSX transcriptional activity was dramatically decreased (Supplementary Fig. 3e). In

addition, *Sirt7* KD in MC3T3-E1 cells decreased OSX transcriptional activity, and overexpression of SIRT7 restored it (Fig. 5f). This effect on the transcriptional activity of OSX was not seen in an inactive SIRT7 mutant (SIRT7[H188Y]) (Fig. 5f), indicating that SIRT7 enzymatic activity was required for the promotion of OSX transcriptional activity.

It has been reported that OSX is modified by phosphorylation, acetylation, and ubiquitination[25–30]. These post-translational modifications regulate its protein stability or DNA binding, but no effect of post-translational modifications on the regulation of its transactivation function in osteoblasts was reported. Therefore, we investigated whether SIRT7-dependent augmentation of the transcriptional activity of OSX was due to the regulation of its protein stability. As shown in Fig. 5g, GAL4DBD-OSX protein levels were not decreased in *Sirt7* KD cells, suggesting that regulation of OSX transactivation activity by SIRT7 is independent of protein stability and DNA binding.

**SIRT7 promotes OSX transactivation by deacylation of K368.** Mouse OSX has a transactivation domain in the N-terminal region (amino acids 27–192) and a zinc finger domain in the C-terminal region (amino acids 293–428)[6]. To map the domain

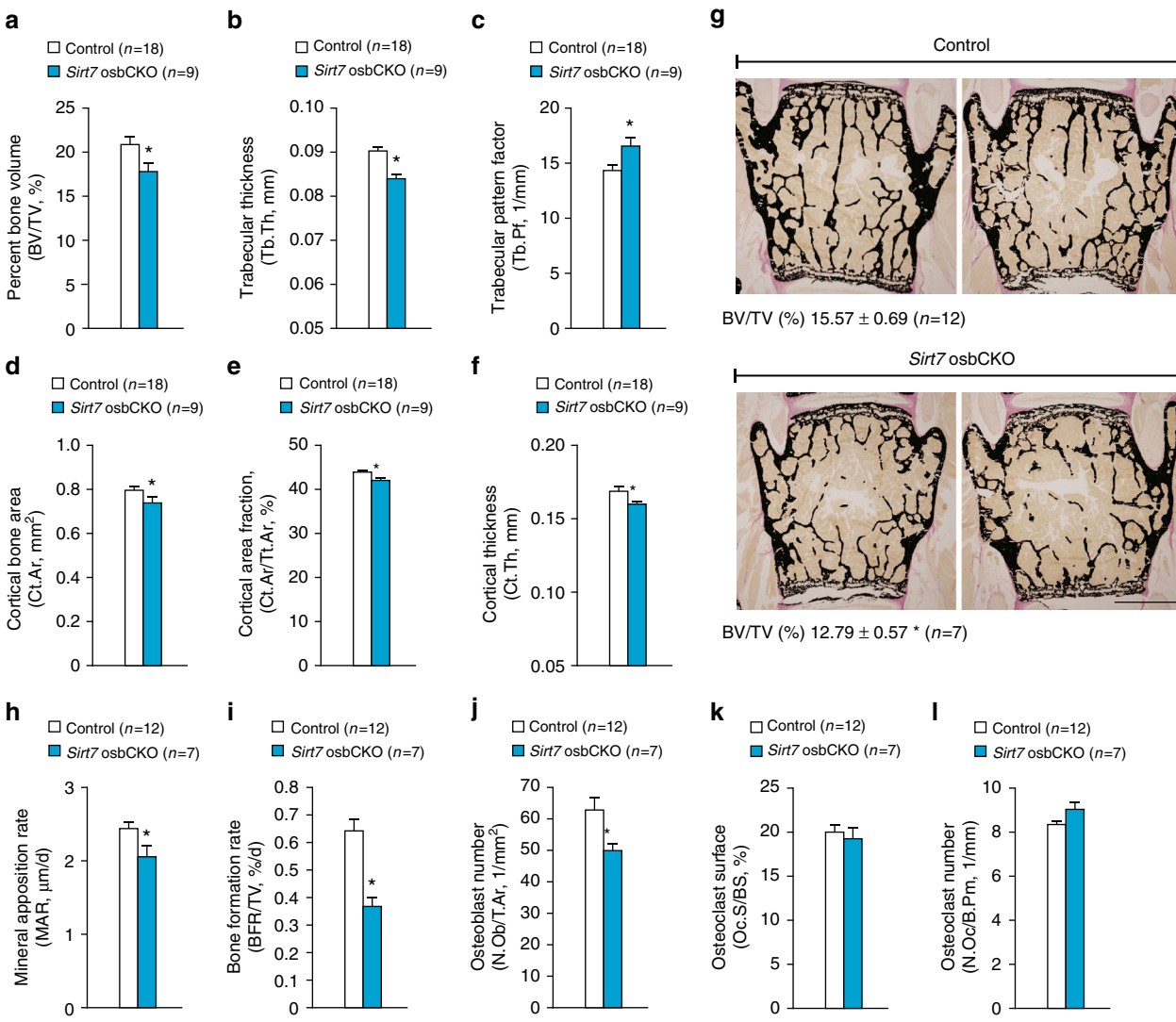

**Fig. 4** Osteoblast-specific *Sirt7* KO mice exhibit reduction of bone mass due to decreased bone formation. **a**–**f** μCT analysis of the distal femur in 15-week-old female *Sirt7* osbCKO mice and control mice. Trabecular BV/TV (**a**), Tb.Th (**b**), Tb.Pf (**c**), Ct.Ar (**d**), Ct.Ar/Tt.Ar (**e**), and Ct.Th (**f**). **g**–**l** Static and dynamic bone histomorphometric analyses of the lumbar spine (L4) in 15-week-old female *Sirt7* osbCKO mice and control mice. Representative von Kossa-stained images and BV/TV (**g**), as well as MAR (**h**), BFR/TV (**i**), N.Ob/T.Ar (**j**), Oc.S/BS (**k**), and N.Oc/B.Pm (**l**). Data are shown as the mean ± SEM. Statistical significance was determined by Student's *t*-test. *$p < 0.05$ vs. control mice. Scale bar, 1 mm

involved in the interaction between OSX and SIRT7, Halo and Halo-SIRT7 proteins were incubated with lysates of HEK293T cells expressing OSX deletion mutants fused with GAL4DBD, and interactions were assessed by the pull-down assay. The C-terminal region of OSX (including the zinc finger domain) showed strong binding to SIRT7, while the N-terminal transactivation domain bound weakly to SIRT7 (Fig. 6a). In MC3T3-E1 cells, *Sirt7* deficiency significantly decreased the transcriptional activity of GAL4DBD-full length OSX (Fig. 6b). It has been reported that the C-terminal region of OSX attenuates its N-terminal transactivation activity through an unknown mechanism[6,31]. Consistent with these reports, deletion of the C-terminal region of OSX led to a marked increase of its trans-activation activity (Fig. 6b). However, there was no reduction of the activity of OSX lacking the C-terminal region in *Sirt7* KD cells, indicating that binding of SIRT7 to the C-terminus of OSX is important for regulating N-terminal transactivation activity (Fig. 6b). Thus, C-terminal deacylation by SIRT7 may promote the N-terminal transactivation activity of OSX.

The C-terminal region of mouse OSX contains ten lysine residues. To investigate the residues targeted by SIRT7, we generated ten different lysine to arginine replacement (KR) mutants and analyzed the transcriptional activity of each OSX mutant. We found that the OSX (K368R) mutant showed a marked increase of transcriptional activity by more than 4-fold compared with WT OSX (Fig. 6c). To assess the influence of OSX protein stability, we treated with a proteasome inhibitor (MG132) before analyzing its transcriptional activity. The OSX (K368R) mutant also showed elevation of transcriptional activity after MG132 treatment (Supplementary Fig. 4a), suggesting that K368 was a strong candidate as a target of SIRT7 and that post-translational modification of OSX at K368 attenuated its N-terminal transactivation activity. Notably, *Sirt7* deficiency did not reduce transcriptional activity of the GAL4DBD-OSX (K368R) mutant in MC3T3-E1 cells (Fig. 6d). Also, over-expression of the OSX (K368R) mutant rescued impairment of mineralization in cultures of *Sirt7* KD MC3T3-E1 cells (Fig. 6e). Interestingly, K368 of mouse OSX is conserved in humans, birds, frogs, and zebrafish (Supplementary Fig. 4b). Taken together, these findings support the assumption that SIRT7 promotes N-terminal transactivation activity of OSX by deacylation of C-terminal K368 (Fig. 6f).

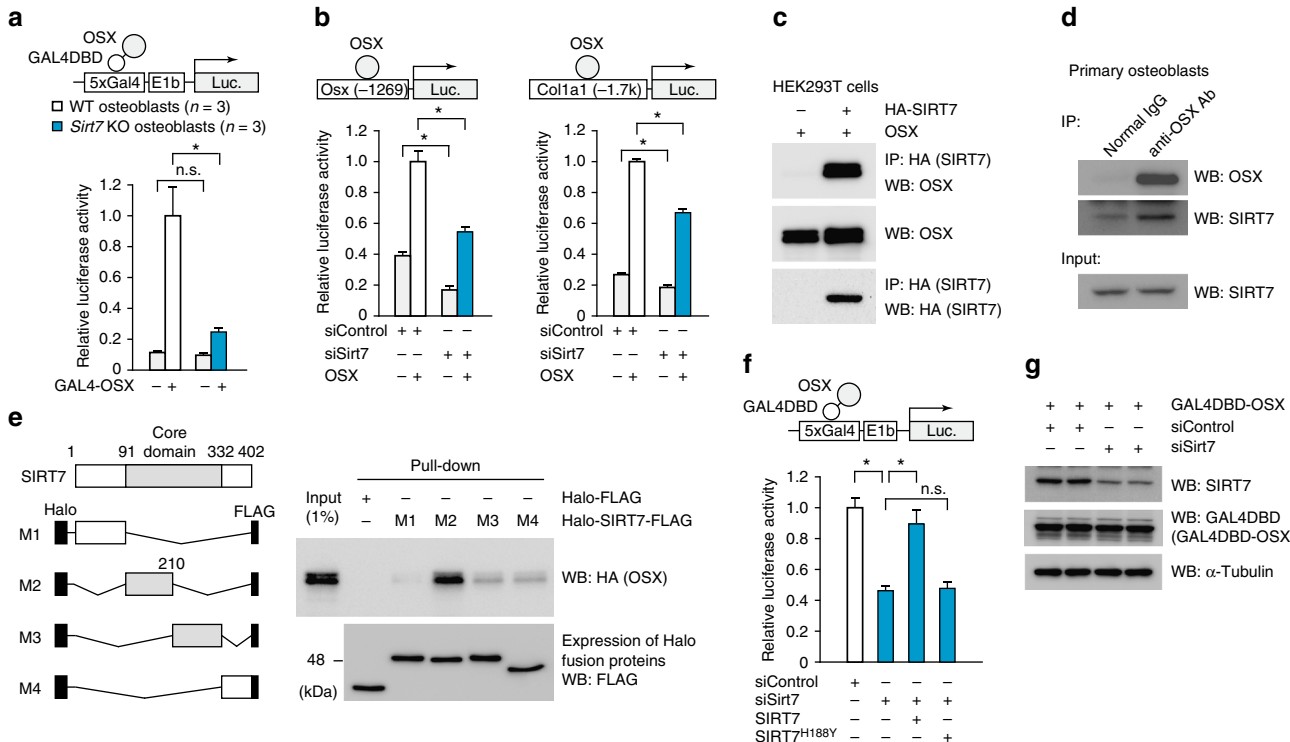

**Fig. 5** SIRT7 interacts with OSX and promotes its transcriptional activity. **a** Transcriptional activity of OSX in *Sirt7* KO and WT osteoblasts. Cells were transfected with the GAL4DBD-OSX expression plasmid and the 5× GAL4-luciferase reporter plasmid, and luciferase activity was determined 24 h after transfection. **b** Transcription of the *Osx* (left) and *Col1a1* (right) enhancer/promoter-driven luciferase reporters in MC3T3-E1 cells. After the indicated siRNA was introduced for 72 h, cells were transfected with the OSX expression plasmid and the indicated luciferase reporter plasmid. The reporter assay was performed 24 h later. *n* = 3 each. **c**, **d** Co-immunoprecipitation assay detecting the interaction between HA-SIRT7 and OSX in HEK293T cells (**c**), and that between endogenous SIRT7 and OSX in primary calvarial osteoblasts differentiated for 6 days (**d**). **e** Mapping the site of interaction between SIRT7 and OSX. Pull-down assay of Halo-SIRT7-FLAG deletion mutants was performed with lysate of OSX-HA overexpressing HEK293T cells. Expression of Halo-FLAG and Halo-SIRT7-FLAG proteins (M1–4 deletion mutants; amino acids 1–90 (M1), 91–210 (M2), 211–332 (M3), and 333–402 (M4)) was determined by WB. **f** Effect of SIRT7 overexpression on transcriptional activity of OSX. MC3T3-E1 cells were transfected with the indicated siRNA, and 72 h later were transfected with the GAL4DBD-OSX expression plasmid, the 5× GAL4-luciferase reporter plasmid, and SIRT7 or SIRT7$^{H188Y}$ expression plasmid. The reporter assay was performed after 24 h. *n* = 3 each. **g** GAL4DBD-OSX protein level analyzed by WB under the conditions in **h**. WB western blotting; IP, immunoprecipitation. Data are shown as the mean ± SEM. Statistical significance was determined by Student's *t*-test. *$p < 0.05$

**Lysine depropionylation of OSX by SIRT7 and SIRT1.** Because SIRT7 possesses deacetylation activity, we examined whether OSX is acetylated in primary calvarial osteoblasts and MC3T3-E1 cells. First we confirmed that acetylation of histone H3 (treated with NAM) and OSX (cotransfected with p300) could be clearly detected by our western blotting system (Supplementary Fig. 5a, b). Using the same system, we found that acetylation of OSX was barely detectable in primary osteoblasts (Fig. 7a) or in MC3T3-E1 cells stably overexpressing HA-OSX (Supplementary Fig. 5c), and acetylation of OSX was slightly increased by NAM treatment (Fig. 7a, Supplementary Fig. 5c). Because sirtuins can also remove other acyl-lysine modifications, such as propionylation, we next performed western blotting analysis with an anti-body targeting propionylated lysine. In contrast of acetylation, propionylation of OSX was clearly detected in primary cultures of osteoblasts and MC3T3-E1 cells and it was strongly enhanced by treatment of cells with NAM (Fig. 7a, Supplementary Fig. 5c). We next assessed whether acetylation/propionylation of OSX was regulated by SIRT7. Propionylation of endogenous OSX was enhanced in primary cultures of osteoblasts derived from *Sirt7* KO mice (in vitro) and in the calvariae of *Sirt7* KO mice (in vivo), while acetylation of OSX showed no significant change (Fig. 7b, c). Propionylation of OSX was also enhanced in MC3T3-E1 cells with *Sirt7* KD (Supplementary Fig. 5d). While propionylation of OSX was not detected in mouse embryonic fibroblasts (MEF), addition of sodium propionate (Na-prop) to the culture

medium led to prominent OSX propionylation (Supplementary Fig. 5e). Under these culture conditions, addition of SIRT7 reduced the propionylation of OSX, while inactive SIRT7$^{H188Y}$ had no effect on OSX propionylation (Fig. 7d). These results demonstrated that SIRT7 reduced propionylation, but not acetylation, of OSX in osteoblasts.

To identify the sites of propionylation, we performed mass spectrometry (MS) using lysates of HA-OSX-expressing 293T cells treated with NAM and Na-prop. This revealed propionylation of OSX, including K41, 45, 358, and 386 (Supplementary Fig. 6a,b). We also found several sites of lysine acetylation, even though the signal detected by western blotting was very weak. However, we could not identify the actual modification of K368 because repeated MS analysis failed to detect the corresponding peptide. OSX with a triple KR mutation (K41R/K45R/K46R) in the N-terminal transactivation domain showed less propionylation than WT OSX (Fig. 7e), demonstrating that at least one of these lysine residues was propionylated in OSX. Finally, we found much less propionylation of the OSX (K368R) mutant than WT OSX in *Sirt7* KO MEFs treated with Na-prop, and showed that SIRT7 did not decrease propionylation of OSX (K368R) (Fig. 7f). These results suggested that SIRT7-mediated deacylation of K368 facilitated the N-terminal depropionylation of OSX.

Since our investigations indicated that SIRT7 increases the transactivation activity of OSX and reduces its propionylation, we

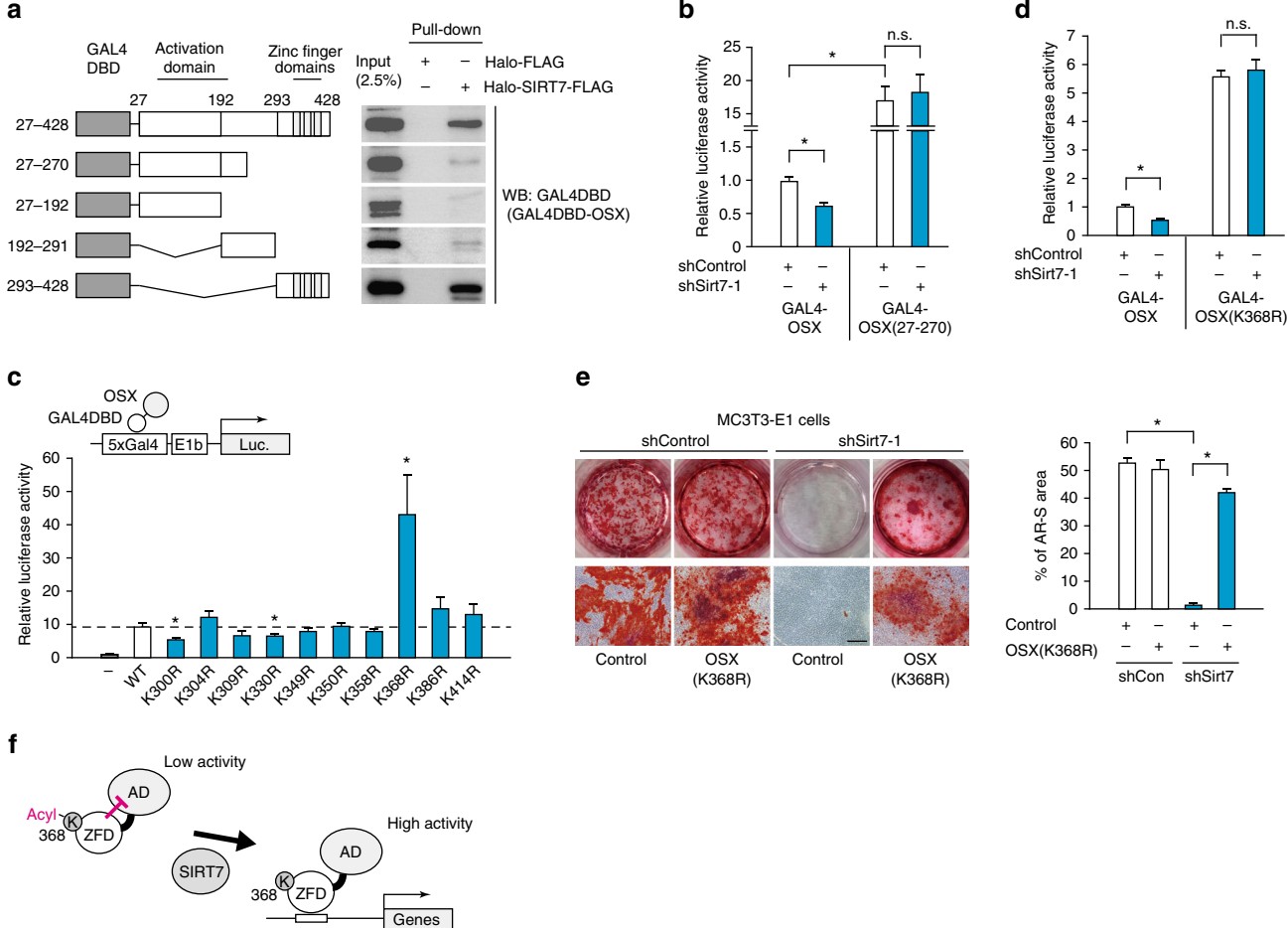

**Fig. 6** SIRT7 increases transactivation activity of OSX by deacylation of lysine 368. **a** Mapping the site of the interaction between OSX and SIRT7. Halo-SIRT7-FLAG pull-down assay with lysates of HEK293T cells expressing the indicated OSX deletion mutants fused with GAL4DBD. **b** Effect of *Sirt7* deficiency on N-terminal transactivation activity of OSX in MC3T3-E1 cells. After the indicated shRNA was introduced, cells were transfected with the GAL4DBD-OSX or GAL4DBD-OSX (27–270) expression plasmid, as well as the 5× GAL4-luciferase reporter plasmid. Luciferase activity was determined at 24 h after transfection (*n* = 6 each). **c** Transactivation activity of OSX KR mutants in MC3T3-E1 cells. Cells were transfected with the GAL4DBD-OSX expression plasmid (WT) or the indicated GAL4DBD-OSX expression plasmid (KR mutants), as well as the 5× GAL4-luciferase reporter plasmid. Luciferase activity was determined after 24 h (*n* = 6 each). **d** Effect of *Sirt7* deficiency on transactivation activity of OSX (K368R) mutants in MC3T3-E1 cells. After the indicated shRNA expression vectors were introduced, cells were transfected with the GAL4DBD-OSX or GAL4DBD-OSX (K368R) expression plasmid, as well as the 5× GAL4-luciferase reporter plasmid. Luciferase activity was determined after 24 h (*n* = 6 each). **e** Effect of OSX (K368R) overexpression in *Sirt7* knockdown MC3T3-E1 cells. The indicated shRNA was introduced into MC3T3-E1 cells, followed by transfection with each OSX (K368R) expression plasmid or empty plasmid. Then cells were cultured for 30 days in differentiation medium. Representative Alizarin Red S-stained images (left) and Alizarin Red S staining area ratio (right). **f** Proposed model for regulation of OSX transactivation activity by SIRT7 through deacylation of lysine 368. See the Discussion section for details. WB western blotting. Data are shown as the mean ± SEM. Statistical significance was determined by Student's *t*-test. *p < 0.05 vs. GAL4DBD-WT OSX (**c**). Scale bar, 400 μm

postulated that lysine propionylation of OSX might inhibit transactivation. We found that treatment of MC3T3-E1 cells with Na-prop inhibited the expression of marker genes for early osteoblastic differentiation, while there was no change in the expression of non-osteoblast related genes, including *beta-2 microglobulin* (*B2m*), *actin, beta* (*Actb*), and *18S ribosomal RNA* (*18S*) (Supplementary Fig. 5f). Na-prop treatment also decreased GAL4DBD-OSX transcriptional activity in a concentration-dependent manner (Fig. 7g). To obtain more direct evidence that lysine propionylation of OSX attenuates its transactivation activity, we performed a luciferase assay with transfection of recombinant propionylated OSX protein. As shown in Fig. 7h, propionylated OSX (purified from OSX-overexpressing *Sirt7* KO MEFs treated with Na-prop) showed lower transcriptional activity than underpropionylated OSX. Taken together, these findings suggested that propionylation of OSX attenuated its transactivation activity.

Recent studies have shown that SIRT1 possesses depropionylation activity, while SIRT6 (another nuclear sirtuin) does not[32]. Therefore, we examined whether SIRT1 caused depropionylation of OSX and whether SIRT7-facilitated such depropionylation. We found that SIRT1 interacted with full-length OSX and also with its N-terminal activation domain (Supplementary Fig. 7a, Fig. 8a). SIRT1 reduced lysine propionylation in OSX, and the combination of SIRT1 and SIRT7 nearly abolished it (Fig. 8b). These results suggested that SIRT1 could be involved in the depropionylation of OSX. Consistent with these findings, SIRT1 and SIRT7 synergistically promoted the transcriptional activity of OSX (Fig. 8c). Moreover, SIRT1 more strongly enhanced the transcriptional activity of OSX (K368R) than that of WT OSX, further supporting the cooperative effect of SIRT7 and SIRT1 (Fig. 8d). In conclusion, the present findings suggested that SIRT7-mediated deacylation of OSX K368 facilitates depropionylation by SIRT1, and thereby

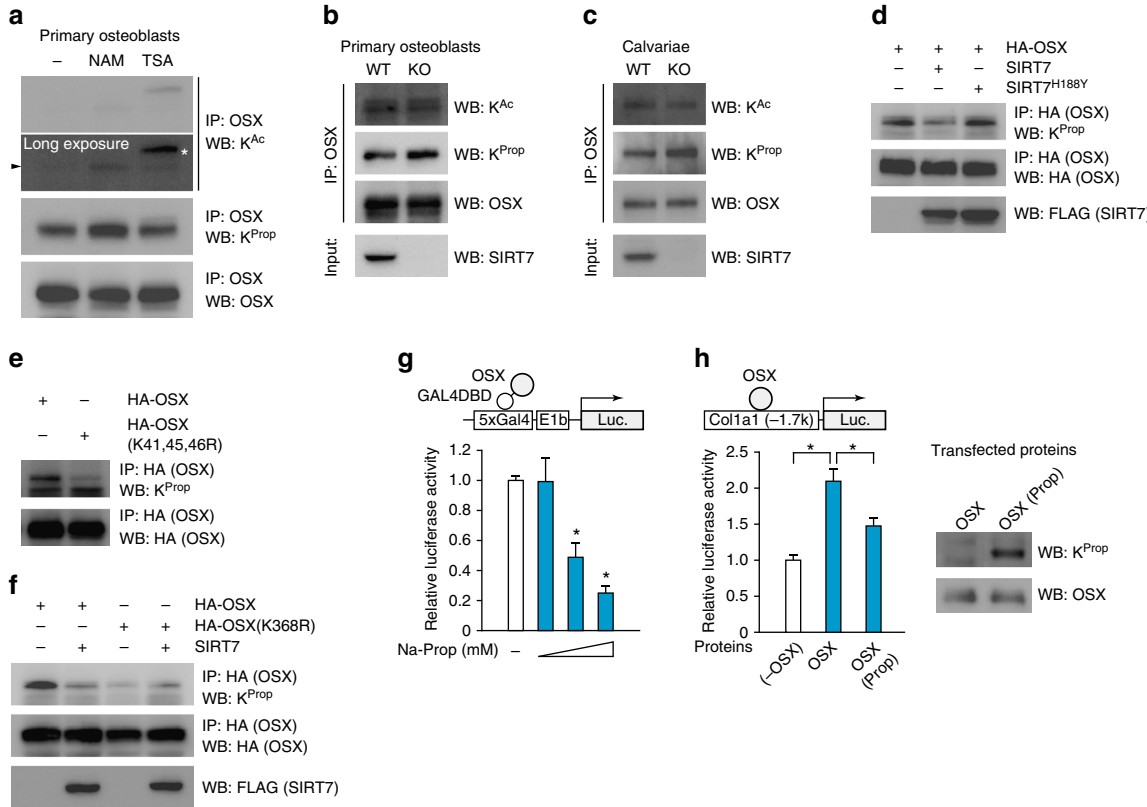

**Fig. 7** SIRT7 regulate OSX lysine propioylation. **a** Detection of acetylated or propionylated OSX. Calvarial osteoblasts differentiated for 6 days were treated with 10 mM NAM for 24 h or with 1 μM TSA for 6 h. After immunoprecipitation, acetylated and propionylated OSX were detected by WB. A long exposure time was required to detect acetylated OSX. Arrowhead and asterisk indicate OSX and non-specific protein, respectively. **b,c** Effect of Sirt7 deficiency on acetylation and propionylation of OSX. Protein lysates of osteoblasts differentiated from bone marrow mesenchymal stem cells (**b**) or of calvariae (**c**) were subjected to immunoprecipitation, after which acetylated and propionylated OSX were detected by WB. **d** Effect of SIRT7 overexpression on propionylation of OSX. Sirt7 KO MEF were transfected with the 3× HA-OSX expression plasmid and the SIRT7 or SIRT7[H188Y] expression plasmid, followed by treatment with 50 mM Na-Prop for 16 h. Propionylation of OSX was assessed by immunoprecipitation and WB. **e, f** Detection of propionylated OSX-mutants. MEF were transfected with the 3× HA-OSX or 3× HA-OSX (K41,45,46R) expression plasmid, after which cells were treated with 50 mM Na-Prop for 16 h (**e**). The 3× HA-OSX or 3× HA-OSX (K368R) expression plasmid combined with the SIRT7 or empty expression plasmid were transfected into Sirt7 KO MEF, which were subsequently treated with 50 mM Na-Prop for 16 h (**f**). Propionylation of OSX was assessed by immunoprecipitation and WB. **g** Impact of propionylation on transactivation activity of OSX. MC3T3-E1 cells were transfected with the GAL4DBD-OSX expression plasmid, and then were treated with 10, 30, 50 mM Na-Prop for 24 h. Subsequently, cells were transfected with the 5× GAL4-luciferase reporter and the reporter assay was performed after 24 h ($n = 5$ each). **h** Luciferase assay using recombinant OSX protein. OSX protein was transfected into MC3T3-E1 cells, followed by transfection of the Col1a1 enhancer/promoter-driven luciferase reporter after 2 h. The reporter assay was performed at 18 h after luciferase transfection (left) ($n = 4$ each). Transfected proteins were confirmed by WB (right). WB, western blotting; IP, immunoprecipitation; K[Ac] acetyl lysine, K[Prop] propionyl lysine. Data are shown as the mean ± SEM. Statistical significance was determined by Student's $t$-test. $*p < 0.05$ vs. 0 mM Na-Prop (**g**)

increases the transactivation activity of OSX (Supplementary Fig. 7b).

## Discussion

This study provided evidence that Sirt7 KO mice developed severe osteopenia due to decreased bone formation along with an increase of osteoclasts (Figs. 1 and 2). Deficiency of Sirt7 in osteoblasts significantly decreased osteoblastic differentiation and ossification in vitro (Fig. 3), and osteoblast-specific Sirt7 KO mice exhibited the reduction of bone mass and decreased bone formation without a change of osteoclast numbers (Fig. 4). We also identified the underlying molecular mechanism by demonstrating that SIRT7 increased the transactivation activity of OSX through deacylation of lysine 368 (Figs. 5 and 6). In addition, we found that SIRT7-mediated deacylation of lysine 368 facilitated depropionylation of OSX, and SIRT1 synergistically promoted depropionylation of OSX with SIRT7 (Figs. 7 and 8). These findings indicate that SIRT7/

SIRT1-mediated deacylation of lysine residues is required for full activation of OSX, allowing these enzymes to regulate osteoblast differentiation and bone formation. To our knowledge, this is the first report concerning a role of SIRT7 in bone turnover and metabolism.

Although we demonstrated that SIRT7 increases the transactivation activity of OSX through deacylation of lysine 368, we could not identify the actual modification of this lysine by MS analysis. When we investigated whether recombinant SIRT7 was able to depropionylate lysine 368 using OSX K368 propionyl peptide, we could not detect depropionylation activity even in the presence of nucleic acid, an activator of SIRT7[33], while we clearly observed deacetylation of H3 K18 acetyl peptide by SIRT7 (Supplementary Fig. 8a, b). Furthermore, acetylation of OSX was unchanged in the calvariae of Sirt7 KO mice (Fig. 7c). Thus, it is unlikely that SIRT7 either deacetylates or depropionylates lysine 368. Recent studies have shown that SIRT7 also has desuccinylase activity and defatty-acylase activity (removing myristoylation)[16,33]. Further investigation will be necessary to

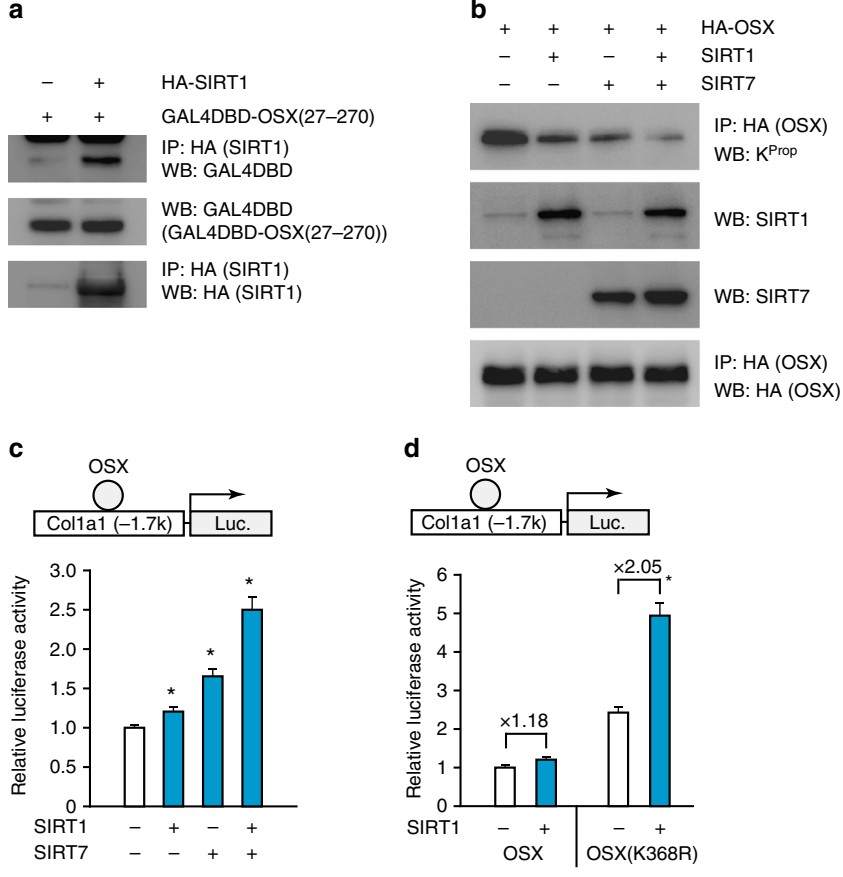

**Fig. 8** SIRT7 and SIRT1 regulate OSX transactivation activity through lysine deacylation. **a** Co-immunoprecipitation assay to assess the interaction between HA-SIRT1 and GAL4DBD-OSX (27–270) in HEK293T cells. **b** Effect of SIRT7 and SIRT1 overexpression on propionylation of OSX. *Sirt7* KO MEF were transfected with the 3× HA-OSX expression plasmid, as well as the SIRT7 or SIRT1 expression plasmid, followed by treatment with 50 mM Na-Prop for 16 h. Then propionylation of OSX was assessed by immunoprecipitation and WB. **c** Effect of SIRT7 and SIRT1 overexpression on transactivation activity of OSX. *Sirt7* KO MEF were transfected with the 3× HA-OSX expression plasmid, as well as the SIRT7 or SIRT1 expression plasmid, followed by treatment with 10 mM Na-Prop for 24 h. Then the cells were transfected with the *Col1a1* enhancer/promoter-driven luciferase reporter plasmid, and the reporter assay was performed after 24 h ($n = 6$ each). **d** Effect of SIRT1 on transactivation activity of the OSX (K368R) mutant. *Sirt7* KO MEF were transfected with the 3× HA-OSX or 3× HA-OSX (K368R) expression plasmid, as well as the SIRT1 expression plasmid, followed by treatment with 30 mM Na-Prop for 24 h. Then the cells were transfected with the *Col1a1* enhancer/promoter-driven luciferase reporter plasmid, and the reporter assay was performed after 24 h ($n = 6$ each). WB, western blotting; IP, immunoprecipitation. Data are shown as the mean ± SEM. Statistical significance was determined by Student's *t*-test. *$p < 0.05$ vs. without SIRT1 and SIRT7 (**c**)

clarify the actual modification of lysine 368 for a better understanding of the molecular mechanism by which SIRT7 regulates the transactivation activity of OSX.

In the present study, we demonstrated synergistic depropionylation of OSX by SIRT1 and SIRT7 (Fig. 8b), resulting in activation of OSX transactivation (Fig. 8c). SIRT1 has already been reported to modulate bone formation by osteoblasts. In MSC-specific *Sirt1* KO mice, it was reported that SIRT1 regulates osteoblastic differentiation of MSCs by deacetylation of β-catenin[12]. Depletion of SIRT1 in osteoblast progenitors employing *Osx-Cre* mice led to a decrease in cortical bone thickness associated with decreased bone formation, resulting from increased sequestration of β-catenin by acetylated-FoxOs[10]. In addition to these mechanisms, our present findings suggest that SIRT1 positively regulates osteoblast differentiation by modulating the propionylation of OSX.

Our findings suggested that reduced bone formation in *Sirt7* KO mice was mainly dependent on OSX. However, some of our data were not in line with published findings about OSX-deficient mice. First, we found that *Sirt7* deficiency was associated with a decrease of *Runx2* expression in MC3T3-E1 cells (Fig. 3g) and in the femur (Fig. 2e, Supplementary Fig. 1m). It was previously reported that

*Runx2* expression is normal in *Osx* KO mice, indicating that Runx2 is upstream of OSX[34]. Baek et al. reported an increase of *Runx2* expression in the long bones of *Osx^flox/−*;*Col1a1-Cre* mice compared with *Osx^flox/+*;*Col1a1-Cre* mice, although the mechanism involved was unclear[35]. Sirtuins have multiple substrates and regulate several intracellular signaling pathways, so we cannot exclude the possibility that SIRT7 also regulates osteoblast differentiation by acting on factors other than OSX, e.g., SIRT7 may elevate *Runx2* expression at a certain stage of osteoblast differentiation. In the future, chromatin immunoprecipitation sequencing (ChIP-seq) will provide further information about the role of SIRT7 in the regulation of osteoblast functions. Second, we found that *Sirt7* deficiency did not affect the proliferation of primary calvarial osteoblasts (Fig. 3b). Previous studies have shown that calvarial osteoblasts from *Osx* KO mice grow faster than WT cells, and calvarial BrdU incorporation was found to be greater in *Osx*-null embryos than in WT embryos[36]. It is generally considered that the canonical OSX pathway involves binding to GC-box DNA elements to regulate the transcription of target genes. Recently, Hojo et al. reported that OSX acts as a transcriptional coactivator in Dlx-containing regulatory complexes bound to AT-rich motifs[24]. OSX can also form complexes with other transcriptional factors[37]. Here

we demonstrated that SIRT7 regulated OSX transcriptional activity mediated via GC-box DNA elements, but did not seem to affect its Dlx coactivator function. Taken together, it can be suggested that SIRT7 partially activates transactivation by OSX, so that OSX functionality is not completely abolished in *Sirt7* KO osteoblasts. Zhang et al. have suggested that disruption of DNA binding by Tcf1, a partner of β-catenin, due to interaction with OSX is at least partly responsible for OSX-mediated inhibition of osteoblast proliferation[36]. Accordingly, the SIRT7-dependent transactivation activity of OSX may not have an important role in osteoblast proliferation.

We found that osteoclast numbers in the lumbar spine were increased in *Sirt7* KO mice compared with WT controls (Supplementary Fig. 1n–p), but SIRT7 was not essential for osteoclastogenesis in our cell culture system (Supplementary Fig. 2). What is the reason for this discrepancy? Osteoclasts are tissue-specific multinucleated macrophages that differentiate from GMP, which arise from common myeloid progenitors (CMP). The cells used in our osteoclastogenesis assay were monocytes/macrophages, raising the possibility that SIRT7 is not essential for differentiation of macrophages to osteoclasts, but is required for differentiation of hematopoietic stem cells to CMP or differentiation of CMP to GMP. Indeed, Mohrin et al. reported that SIRT7 is involved in the maintenance of hematopoietic stem cells and that myeloid-biased differentiation was apparent in *Sirt7* KO mice[38]. Alternatively, SIRT7 may regulate modulators of osteoclastogenesis derived from sources other than osteoblasts or osteoclasts.

Acetylation of lysine is a well-known post-translational modification involved in a wide variety of cellular processes[39]. In contrast, propionylation of lysine is a modification that was only recently identified in mammalian cells, and has been shown to occur in histones, p53, p300, and CREB-binding protein (CBP)[40,41]. Although propionylation has been characterized in multiple proteins and organisms, the biological effects of lysine propionylation are poorly understood, as are the differences between propionylation and acetylation. In this study, we demonstrated that OSX undergoes lysine propionylation, leading to the reduction of its transactivation activity (Fig. 7). We could not identify differences in transactivation activity between propionylation and acetylation of OSX, because we were unable to prepare acetylated (but not propionylated) recombinant OSX for the luciferase assay. In contrast of functional difference, we demonstrated that these two closely related acylation marks were differentially regulated by SIRT7. Propionylation of endogenous OSX, but not its acetylation, was increased in primary osteoblasts and calvariae obtained from *Sirt7* KO mice (Fig. 7b, c). Further investigations will be needed to define the molecular mechanisms involved. It is possible that SIRT7-facilitated deacetylation of lysine residues in OSX affects a limited number of several acetylated lysine residues, so that small changes due to SIRT7 are masked in whole.

Propionic acidemia (PA) is a rare autosomal recessive disorder characterized by the accumulation of propionic acid due to deficiency of propionyl-CoA carboxylase[42,43]. Adults with PA have an increased risk of osteoporosis or osteopenia, but the reason is unknown. Elevated global propionylation of lysine has been observed in fibroblasts from PA patients[44]. Based on our results, propionylation of OSX may be elevated in PA osteoblasts, and may at least partly contribute to the pathophysiology of osteoporosis in this disease.

In conclusion, we provided evidence that acylation of lysine 368 in OSX is a post-translation modification involved in regulating transactivation. The importance of the role of SIRT7 as an ERASER of lysine acylation in regulating osteoblast differentiation was also clarified. Our findings will hopefully provide a stepping-stone for future studies on the role of lysine non-acetyl

acylation in bone metabolism. Further studies of acyltransferases as WRITERS, as well as ERASERS, and the influence on acyl-CoA metabolism will broaden our understanding of the biological significance of lysine acylation. Finally, SIRT7 could be a potential target for the treatment of osteoporosis.

## Methods

**Mice.** All experimental procedures were approved by the Kumamoto University Ethics Review Committee for Animal Experimentation (Approval ID: A27-024, A29-001). All mice were maintained on a 12-h light/dark cycle with access to regular chow and water ad libitum, unless otherwise specified. These experiments were conducted according to the guidelines of the Institutional Animal Committee of Kumamoto University. Generation of *Sirt7* KO mice was accomplished as reported previously[19]. *Sirt7* KO mice were backcrossed for five generations with C57/BL6J mice (CLEA Japan Inc.). Only the heterozygotes ($Sirt7^{+/−} \times Sirt7^{+/−}$) were bred and littermates (WT and *Sirt7* KO mice) were used for these studies. There was no apparent increase of embryonic lethality, postnatal death, or growth retardation in *Sirt7* KO mice, as shown in two previous independent studies[17,18]. *Sirt7* osbCKO mice were generated by intercrossing the progeny of crosses between *Sirt7*$^{flox/flox}$ mice[17] (harboring LoxP sites in introns 5 and 9) and transgenic mice expressing Cre recombinase under the control of the α1(I)-collagen promoter (Col1a1-Cre), which drives *Cre* expression in osteoblasts and odontoblasts after embryonic day 14.5[45]. *Sirt7*$^{flox/flox}$ littermates were used as controls. Genotyping of mice was done by PCR using primer sequences that are available upon request. Serum Ca and P levels were measured by The Institute of Resource Development and Analysis (IRDA) at Kumamoto University.

**Plasmids.** GAL4DBD-OSX expression plasmids [pSG C22 (27–428), pSG C22 (27–270), pSG C22 (27–192), pSG C22 (192–291), and pSG C22 (293–428)], a 3× HA-OSX expression plasmid (pEx3.1Osx), and a 5× GAL4-luciferase reporter plasmid (pG5E1b-luc) were generated as described previously[6]. Various KR mutations of OSX were introduced by using a KOD-Plus-Mutagenesis Kit (TOYOBO) and were verified by sequencing. Then each mutant was cloned into pBIND (Promega) for GAL4DBD fusion, pTriEx-1.1 (Novagen) for 3× HA-OSX expression, and the pMXs-Neo Retroviral Vector (Cell Biolabs, Inc.). The mouse osterix promoter (−1269/+91) driven luciferase reporter plasmid [pGL4.18 osx promoter (−1269/+91)], RUNX2 expression plasmid (pcDNA3-RUNX2), and reporter plasmid (p6 × OSE2-luc) have been described previously[22]. Mouse Col1a1 promoter-driven luciferase reporter [Col1a1 promoter (−1700/+100)] was amplified by PCR and was cloned into pGL4.21 (Promega). A double-stranded oligonucleotide for the 6× AT-rich motif (TAATTATAGGTT) was introduced into pGL4.28 (Promega). The full-length coding sequences of mouse OSX (UniProt number: Q8VI67), Dlx5, and SIRT1 were amplified by PCR and verified by sequencing, followed by cloning into pcDNA3.1, pcDNA3-HA, and pcDNA3-FLAG. To generate OSX-HA expression plasmids, a double-stranded oligonucleotide encoding the HA epitope was introduced into the frame at the C-terminal end of the pcDNA3.1-OSX. *Escherichia coli* Halo fusion SIRT7 expression plasmids [pFN18A-Halo-FLAG, pFN18A-Halo-SIRT7-FLAG, and pFN18A-Halo-SIRT7 (FL/M1/M2/M3/M4)-FLAG] and mammalian SIRT7 expression plasmids (pcDNA3.1-FLAG-SIRT7, pcDNA3.1-HA-SIRT7, and pcDNA3-FLAG-SIRT7$^{H188Y}$) were generated as described previously[46]. pCMVβ-p300-myc was a gift from Tso-Pang Yao (Addgene plasmid # 30489).

**Cell culture.** A murine preosteoblast cell line (MC3T3-E1; RIKEN BRC, Japan) was cultured in α-minimal essential medium (α-MEM) with 10% fetal bovine serum (FBS). A human osteosarcoma cell line (U2OS; ATCC, VA) was cultured in McCoy's 5A (Modified) Medium with 10% FBS. HEK293T cells (Clontech) and a murine macrophage-like cell line (RAW264.7; ATCC) were cultured in Dulbecco's modified Eagle's medium (DMEM) with 10% FBS. A murine embryonic mesenchymal cell line (10T1/2; RIKEN BRC) was cultured in basal Eagle's medium (BME) with 10% FBS. To obtain OSX-overexpressing MC3T3-E1 cells, Plat-E cells were transfected with pMXs-3× HA-OSX, pMXs-3× HA-OSX (K368R), or the negative control pMXs vector using JetPRIME transfection reagent (Polyplus, NY). The retroviruses were infected to MC3T3-E1 cells, and then the infected cells were selected by culture with puromycin (5 μg ml⁻¹). Primary calvarial osteoblasts were harvested as described previously[47]. Briefly, calvariae from neonatal mice 3–4 days old were digested for 1 h with 0.1% collagenase type II (Worthington Biochemical Corporation, NJ) and 0.06% trypsin (Gibco BRL, Life Technologies) in α-MEM while shaking gently at 37 °C. Then the cells were collected by centrifugation and seeded into culture dishes in α-MEM with 5% FBS. Primary cultures of osteoblasts differentiated from bone marrow mesenchymal stem cells were prepared as follows. The femoral and tibial bone marrow was flushed out with medium (StemXVivo Mesenchymal Stem Cell Expansion Media, R&D system, MN), and the resulting cell suspension was placed in a 10 cm dish. After incubation for 5 days under hypoxic conditions, cells were subcultured in a new 10 cm dish. Confluent cells were cultured for 3 days in α-MEM with 5% FBS, followed by incubation for 4 days in differentiation medium supplemented with 10 mM β-glycerophosphate, 50 μg ml⁻¹ ascorbic acid, and 10 ng ml⁻¹ recombinant human BMP-2 (rhBMP-2, a

gift from Osteopharma Inc., Osaka, Japan). Primary MEF were isolated as follows. Dissected mouse embryos of WT and *Sirt7* KO mice (E13.5) were minced and incubated in 0.05% trypsin-EDTA for 20 min in a 37 °C incubator. Then cells were collected by centrifugation and seeded into culture dishes containing fresh MEF medium (DMEM with 25 mM glucose, 1.0 mM pyruvate, 10% (v/v) FBS, and 0.1% (v/v) penicillin/streptomycin). To study mineralization, confluent cells were cultured for 3 days in α-MEM with 5% FBS, followed by incubation in differentiation medium supplemented with 10 mM β-glycerophosphate and 50 μg ml$^{-1}$ ascorbic acid. The medium was changed every 3 days. After culture for 10 days (for primary osteoblasts) or 30 days (for MC3T3-E1 cells), the cells were subjected to Alizarin red S staining. The osteoblast proliferation assay was performed using WST-1 (Roche) according to the manufacturer's instructions, while the osteoblast-free osteoclast differentiation assay was done as described previously[45]. Briefly, bone marrow cells ($3 \times 10^5$ cells per cm$^2$) harvested from the femurs of mice aged 6–8 weeks were cultured for 2 days in α-MEM with 10% FBS and 10 ng ml$^{-1}$ human M-CSF (R&D Systems), and then differentiation into osteoclasts was achieved by incubation with 50 ng ml$^{-1}$ human RANKL (Peprotech) and M-CSF for 4 days. Subsequently, osteoclast differentiation was evaluated by TRAP staining. For differentiation, RAW264.7 cells ($5 \times 10^3$ cells per cm$^2$) were cultured in DMEM with 10% FBS and 50 ng ml$^{-1}$ RANKL for 5 days. The medium was changed daily. The co-culture osteoclast-forming assay was performed as previously described[47]. Briefly, primary osteoblasts ($5 \times 10^3$ cells per cm$^2$) were cultured alone for 1 day and then were co-cultured for 15 days with splenocytes ($1 \times 10^5$ cells per cm$^2$) obtained by mincing spleens in α-MEM with 10% FBS and $1 \times 10^{-8}$ M 1,25-dihydroxyvitamin D$_3$.

**μCT and bone histomorphometric analyses**. Mice were injected with Calcein (Sigma) twice (days 1 and 4) and then were sacrificed on day 6. The femur was harvested, fixed, and scanned using a Skyscan 1076 μCT scanner (Bruker Corporation) according to the guidelines of American Society for Bone and Mineral Research. Structural bone parameters were determined with CTAn analysis software (Skyscan). For bone histomorphometric analysis, undecalcified vertebrae were embedded in methylmethacrylate. Serial sections were stained using von Kossa, Toluidine blue, and TRAP, after which static and dynamic bone parameters were measured using the Osteomeasure Analysis System (Osteometrics) as described previously[45].

**Gene expression analysis**. After total RNA was extracted by using Sepasol RNA I super reagent (Nacalai Tesque, Japan), qRT-PCR was performed with SYBR Premix Ex Taq II (RR820A, TaKaRa) and an ABI 7300 thermal cycler (Applied Biosystems, CA). For extraction of total RNA from femur, soft tissues and epiphyses were removed, and diaphyses were flushed with PBS to remove bone marrow. The relative expression of each gene was normalized to that of *TATA box binding protein* (*Tbp*). Primer sequences are listed in Supplementary Table 1.

**RNA interference experiments**. For transient knockdown of *Sirt7*, transfection of Sirt7 siRNA (FlexiTube siRNA Mm_Sirt7_5; Qiagen) was performed with HiPerFect transfection reagent (Qiagen) according to the manufacturer's protocol. The control was AllStar Negative Control siRNA (Qiagen). For stable knockdown of *Sirt7*, specific shRNA sequences targeting mouse *Sirt7* were designed using the Clontech RNAi target sequence selector (shSirt7-1: 5′-AGATTATCGGGGTCC-TAAT-3′ and shSirt7-2: 5′-ACATGAGCATCACCCGTTT-3′). Oligonucleotides were synthesized and cloned into the pSIREN-RetroQ retroviral shRNA expression vector (Clontech). Then pSIREN-RetroQ-Sirt7 and the negative control vector (pSIREN-RetroQ) were transfected into Plat-E cells using JetPRIME transfection reagent (Polyplus, NY). Subsequently, MC3T3-E1 and RAW264.7 cells were infected with the retroviruses and selected by incubation with puromycin (5 μg ml$^{-1}$).

**Western blotting and antibodies**. Total lysates of cells and tissues were obtained by lysis in RIPA buffer (50 mM Tris–HCl (pH 8.0), 150 mM NaCl, 0.1% SDS, 1% NP-40, 5 mM EDTA, and 0.5% sodium deoxycholate) with a protease inhibitor cocktail (Nacalai Tesque). For western blotting, proteins were separated by SDS polyacrylamide gel electrophoresis and transferred to a polyvinylidene fluoride (PVDF) membrane (Immobilon-P; Millipore, MA), which was probed with the primary antibodies. After incubation with the secondary antibodies, proteins were visualized by using Chemi-Lumi One Super (Nacalai Tesque) and a LAS-1000 imaging system (Fuji Film, Japan). The primary antibodies were anti-SIRT7 rabbit monoclonal antibody (clone D3K5A, #5360, Cell Signaling Technology, ×3000 dilution in 5% BSA), anti-histone H3 antibody (sc-10809, Santa Cruz Biotechnology, Inc., ×5000 dilution in 3% skim milk), anti-β-actin antibody (clone AC-15, #A5441, Sigma–Aldrich, ×2000 dilution in 3% skim milk), anti-DYKDDDDK (FLAG) tag antibody (clone 1E6, #018-22381, Wako Pure Chemical Industries, Ltd., ×2000 dilution in 3% skim milk), anti-HA tag antibody (clone 3F10, #11867423001, Roche Applied Science, ×1000 dilution in 5% BSA), anti-c-Myc antibody (clone 9E10, #017-21871, Wako Pure Chemical Industries, Ltd., ×3000 dilution in 5% BSA), anti-osterix antibody (ab22552, Abcam, ×1000 dilution in 3% skim milk), anti-GAL4DBD antibody (G3042, #073M4761, Sigma–Aldrich, ×1000 dilution in 5% BSA), anti-α-tubulin antibody (clone EP1332Y, #1878-1,

Epitomics, Inc., ×3000 dilution in 3% skim milk), anti-SIRT1 antibody (#2028, Cell Signaling Technology, ×1000 dilution in 5% BSA), anti-acetylated-lysine antibody (#9441, Cell Signaling Technology, ×1000 dilution in 5% BSA), and pan anti-propionyllysine rabbit polyclonal antibody (#PTM201, LOT#Z338F112P3, PTM Biolabs Inc., ×1000 dilution in 5% skim milk). Uncropped scans of all immunoblots are shown in Supplementary Fig. 9.

**Luciferase assay**. Cells were transfected with various plasmid DNAs using jet-PRIME transfection reagent (Polyplus). Indicated time after transfection, cells were lysed and assayed using Firefly luciferase and Renilla luciferase substrates in the Dual-Luciferase Reporter Assay System (Promega). Firefly luciferase activity was normalized to Renilla luciferase activity in the same cell extract to correct for variation in transfection efficiency. The total amount of transfected DNA was kept constant by the addition of empty vectors. The pRL-TK Renilla luciferase plasmid was used as an internal control, except pBIND system. The pBIND plasmid not only expresses GAL4DBD fusion protein, but also expresses Renilla luciferase under the control of the SV40 promoter, which allows the user to normalize for differences in transfection efficiency. For transfection of recombinant protein, recombinant propionylated or underpropionylated OSX protein was purified from OSX-transfected *Sirt7* KO MEF treated or untreated with 50 mM Na-prop, respectively. Protein (2 μg well$^{-1}$ (24-well plate)) was transfected with Xfect Protein Transfection Reagent (Clontech Laboratories, Inc.) according to the manufacturer's protocol.

**HaloTag pull-down assay**. Either pFN18A-Halo-FLAG, pFN18A-Halo-SIRT7-FLAG, or pFN18A-Halo-SIRT7 (M1/M2/M3/M4)-FLAG was transfected into *E. coli* K12 (Single Step (KRX) Competent Cells, Promega). *E. coli* was pre-cultured overnight in 5 ml of LB medium with ampicillin. Then the pre-culture was diluted to 1:1000 using the same medium and culture was continued at 37 °C until OD600 reached 0.5. After further culture at 20 °C until OD600 reached 0.7, 20% rhamnose was added at a 1:200 dilution and incubation was done overnight. Following centrifugation at 1400×g, the pellet was resuspended in Halo purification buffer (50 mM Hepes-KOH (pH 7.4), 150 mM NaCl, 1% NP-40, 1 mM PMSF, and protease inhibitor cocktail (Nacalai Tesque)). Then the cells were disrupted by three freeze–thaw cycles, followed by sonication twice for 20 s each at level 2 (Sonifier-150, Branson). After centrifugation at 6000×g for 10 min, the cleared lysate was incubated with HaloLink Resin (Promega) overnight at 4 °C. After binding, the resin was washed five times with the same Halo purification buffer. Halo fusion proteins (30 μg) immobilized on HaloLink Resin were incubated with 500 μg of cell lysate in pull-down buffer (10 mM Tris–HCl (pH 7.4), 150 mM NaCl, 1% NP-40, 10 mM NaF, 10 mM Na$_4$P$_2$O$_7$, 1 mM PMSF, and protease inhibitor cocktail (Nacalai Tesque)). After incubation overnight at 4 °C for binding, the resins were washed five times with the same pull-down buffer and the bound proteins were separated by SDS-PAGE.

**Proteomic identification of proteins**. IP samples were used in solution and in gel for higher accuracy and precision MS analyses. To prepare the sample in solution for MS, 100 μL of 0.1 M glycine HCl (pH 2.0) was added to the washed resin beads, and the mixture was incubated at room temperature for 5 min with gentle shaking. The supernatant was obtained and neutralized with 10 μL of 10× TBS (0.5 M Tris–HCl, 1.5 M NaCl, pH 7.4). These samples were frozen at −80 °C until use. After cysteine reduction and alkylation by DTT and iodoacetoamido, respectively, the samples were digested with Trypsin/Lys-C Mix (Promega) in the solution containing 50 mM Tris–HCl (pH 8.5) at 37 °C overnight. To stop the reaction, trifluoroacetic acid (TFA) was added to the sample at final concentration of 1%. The samples were desalted using ZipTip C18 pipette tips (Millipore) or styr-enedivinylbenzene (SDB)-StageTip[48], and dissolved with 0.1% TFA in 2% acetonitrile (ACN) for LC-MS analyses. To identify proteins detected in SDS-PAGE, in-gel digestion of proteins was performed according to the previous report[49]. The samples were separated onto SDS-PAGE gels. The gels were fixed with 30% methanol and 7.5% acetic acid and then stained with LavaPurple$^{TM}$ Total Protein Stain (FLUOROtechnics) according to the manufacturer's instructions. After scanning the gel image with a Amersham Typhoon scanner RGB system (GE Healthcare), protein bands of interests were selected with Progenesis software (Nonlinear Dynamics) and marked by an Ettan spot picker (GE Healthcare). The gel pieces manually cut out were washed three times with 50 mM ammonium bicarbonate in 50% (v/v) ACN, dehydrated in 100% (v/v) ACN, and vacuum-dried. For cysteine reduction, 100 μL of 10 mM of DTT in 100 mM ammonium bicarbonate was added to the gel pieces which were incubated at 56 °C for 1 h, and was removed. Following cysteine alkylation, 100 μL of iodoacetoamido in 100 mM ammonium bicarbonate was added, and the gel pieces were incubated at 24 °C for 45 min, and was removed. The gel pieces were washed once with 100 mM ammonium bicarbonate, dehydrated in 100% (v/v) ACN, and vacuum-dried. The sequencing grade modified trypsin (Promega) was added to the gel pieces at a concentration of 50 ng ml$^{-1}$ in 10% (v/v) ACN including 50 mM ammonium bicarbonate, and the mixture was incubated on ice for 30 min followed by the incubation at 37 °C overnight. The trypsinized peptides were sequentially extracted from the gels with 0.1% (v/v) TFA in 30% (v/v) ACN, 0.1% (v/v) TFA in 50% (v/v) ACN, and 0.1% (v/v) TFA in 80% (v/v) ACN, for 5 min each. The extracted

peptides were vacuum-dried and dissolved in 20 µL of 0.1% (v/v) TFA in 2% (v/v) ACN. These samples were desalted with a styrenedivinylbenzene (SDB)-StageTip, dissolved with 0.1% TFA in 2% ACN, and subjected to LC-MS analysis. The peptide samples were subjected a nano-flow reversed-phase LC-MS/MS system (EASY-nLC™ 1200 System coupled to an Orbitrap Fusion Tribrid Mass Spectrometer; Thermo Fisher Scientific, San Jose, CA) with a nanospray ion source in positive mode. Samples were separated with a nano-HPLC C18 capillary column (0.075 × 150 mm, 3 mm) (Nikkyo Technos, Tokyo, Japan). A 60-min gradient was used at a flow rate of 300 nL min$^{-1}$. The spray voltage was 2.2 kV with ion transfer tube temperature 250 °C. MS and MS/MS scan properties were as follows; Orbitrap MS resolution 120,000, MAS scan range 350–1500, isolation window $m/z$ 1.6, and MS/MS detection type was ion trap with a rapid scan rate. The peptide and fragment mass tolerances were 10 ppm and 0.6 Da, respectively.

The $m/z$ values of propionyl-peptides identified by data-dependent MS analysis were listed into inclusion list for targeted mass mode analysis ($m/z$ = 529.8029 VYG$^{304}$K$^{Prop}$ASHLK, 560.7973 DSTTLG$^{41}$K$^{Prop}$GGTK, 572.9647 $^{287}$K$^{Prop288}$K$^{Ac}$PIHSC$^{Cam}$HIPGC$^{Cam}$GK/$^{287}$K$^{Ac288}$K$^{Prop}$PIHSC$^{Cam}$HIPGC$^{Cam}$GK, 620.8126 FTC$^{Cam}$LLC$^{Cam}$S$^{358}$K$^{Prop}$R, 703.6846 THGEPGPGPPPSGP$^{386}$K$^{Prop}$ELGEGR, 744.908 GGT$^{45}$K$^{Prop}$KPYADLSAPK). MS and MS/MS scan properties for targeted mass analysis were performed as follows; Orbitrap MS resolution 120,000, MAS scan range 350–1500, isolation window $m/z$ 1.6, and MS/MS detection type was Orbitrap with resolution 15,000. The peptide and fragment mass tolerances were 10 ppm and 0.02 Da, respectively. K propionylation (K$^{Prop}$), K acetylation (K$^{Ac}$), C carbamidomethylation (C$^{Cam}$).

All MS/MS spectral data were searched against entries for mice in the Swiss-Prot database (v2016-10-05) using the SEQUEST database search program using Proteome Discoverer 2.2 (PD2.2). For variable peptide modifications, propionylation and acetylation of lysine and oxidation of methionine, in addition to carbamidomethylation of cysteine for a fixed modification, were taken into account. Database search results were filtered by setting the peptide confidence value as high (FDR < 1%) for data dependent mass analysis data, and high correlation (XCorr > 1.9) for targeted mass analysis data, respectively.

**Co-immunoprecipitation assays.** HEK293T cells were transfected with the indicated expression plasmids for 24 h using JetPRIME transfection reagent (Polyplus, NY). Primary cultures of calvarial osteoblasts were differentiated for 6 days. Cells were lysed in IP buffer (20 mM Tris–HCl (pH 7.4), 200 mM NaCl, 2.5 mM MgCl$_2$, 0.05% NP-40, 1 mM PMSF, and protease inhibitor cocktail (Nacalai Tesque)) and then were incubated on ice for 30 min. Next, the lysed cells were passed through a 29 G needle (Terumo) 6 times. After centrifugation at 14,000×$g$, the cleared lysates were subjected to immunoprecipitation overnight at 4 °C with anti-HA antibody beads (clone 4B2, Wako Pure Chemical Industries, Ltd.) or anti-OSX-conjugated magnetic beads, which were prepared using 10 µg of OSX antibody (ab22552, Abcam) and 1 mg of beads according to the protocol of the Dynabeads Antibody Coupling Kit (Invitrogen). Then the beads were washed five times with IP buffer, after which proteins were eluted using 2× SDS sample buffer (100 mM Tris–HCl (pH 6.8), 4% SDS, 20% glycerol, and 0.2% bromophenol blue) and were examined by western blotting with the indicated antibodies.

**Immunocytochemistry.** Both the 3× HA-OSX and FLAG-SIRT7 expression plasmids were transfected into MC3T3-E1 cells with JetPRIME transfection reagent (Polyplus). After 24 h, cells were fixed in 10% neutralized formalin and permeabilized with 0.1% Triton X-100/3% BSA/PBS. Monoclonal rat anti-HA antibody (1:1000) and mouse anti-FLAG antibody (1:1000) were used as the primary antibodies, while Alexa Fluor 568 goat anti-rat IgG and Alexa Fluor 488 goat anti-mouse IgG were the secondary antibodies. Immunofluorescence was detected under a laser scanning confocal microscope (FV-1000, Olympus).

**Detection of lysine acetylation and propionylation.** Cells were lysed in IP buffer (20 mM Tris–HCl (pH 7.4), 200 mM NaCl, 2.5 mM MgCl$_2$, 0.2% NP-40, 1 mM PMSF, 10 mM NAM, 1 µM TSA and EDTA-free protease inhibitor cocktail (Nacalai Tesque)), and were disrupted by sonication with two 30-s pulses at an interval of 30 s at 4 °C (Bioruptor UCD-300, CosmoBio). Calvariae harvested from 10-week-old male *Sirt7* KO mice or WT mice were homogenized with a Polytron PT1200 homogenizer (Kinematica AG, Switzerland) in IP buffer at 4 °C. Soluble fractions were separated by centrifugation at 14,000×$g$ for 10 min at 4 °C. Then the cell lysate was mixed with anti-HA antibody beads (Wako) or anti-OSX-conjugated magnetic beads, which were prepared using 10 µg of OSX antibody (ab22552, Abcam) and 1 mg of beads according to the protocol of the Dynabeads Antibody Coupling Kit (Invitrogen), and was stirred for 15 h at 4 °C. After washing with IP buffer (the same composition as above except for 1% NP-40), proteins were eluted with 1× SDS sample buffer (100 mM Tris–HCl pH 6.8, 4% SDS, 20% glycerol and 0.2% bromophenol blue). Finally, acetylation and propionylation of lysine were detected by western blotting with pan anti-Ac-K antibody (Cell Signaling Technology) and an anti-propionyllysine rabbit polyclonal antibody (PTM Biolabs Inc., LOT#Z338F112P3), respectively.

**Preparation of recombinant SIRT7.** pFN18A-Halo-SIRT7 plasmid was transformed into *E. coli* K12 (KRX) cell. A single colony was inoculated with LB medium and the preculture was further cultured and expressed in LB medium. The Halo-SIRT7 protein was purified from the resulting *E. coli* lysate using HaloLink resin (Promega, # G1914). ProTev Plus protease (Promega, #V6101) was used for separation of the SIRT7 protein from Halo-tag.

**In vitro deacetylation and depropionylation assay.** H3 K18Ac (KSTGGKAPR-K$^{Ac}$QLATKAARKS), OSX K368 (RFTRSDHLSKHQRTHGEPGP) and OSX K369Prop (RFTRSDHLSK$^{Prop}$HQRTHGEPGP) peptides were used for the in vitro deacetylation/depropionylation assay. 8 µM of peptides were incubated in 60 µL reaction volume, with or without 3 µg of recombinant SIRT7 with 1 µg nucleic acid containing 1.0 mM NAD in 20 mM Tris–HCL buffer (pH 7.5) and 1 mM DTT for 2 h at 37 °C. The reactions were stopped with 0.1% formic acid and analyzed by LC-MS/MS.

Peptides were analyzed by using Agilent 6460 Triple Quadruple LC/MS (Agilent Technologies, Santa Clara, USA) to perform LC-ESI-MS/MS. LC conditions employed were as follows; column, Agilent Zorbax Eclipse Plus C18 (2.1 × 50 mm) (Agilent Technologies); column temperature, 45 °C; injection volume, 3 µL; mobile phase, A, 0.1% formic acid, B, acetonitrile; gradient (B concentration), 0 min—3%, 10 min—8%, 10.1 min—3%, 15 min—3%; flow rate, 0.2 ml/min. Multiple reaction monitoring parameters were determined were as follows; precursor ion ($m/z$), 695.8, product ion ($m/z$), 84.1, fragmentor voltage (V), 210, collision energy (V), 55, polarity, positive for H3 K18, precursor ion ($m/z$), 709.8, product ion ($m/z$), 84.1, fragmentor voltage (V), 170, collision energy (V), 55, polarity, positive for acetylated H3 K18 (H3 K18Ac), precursor ion ($m/z$), 469.7, product ion ($m/z$), 70.2, fragmentor voltage (V), 90, collision energy (V), 55, polarity, positive for OSX K368, and precursor ion ($m/z$), 480.9, product ion ($m/z$), 70.2, fragmentor voltage (V), 90, collision energy (V), 55, polarity, positive for propylated OSX K368 (OSX K368Prop).

**Statistical analysis.** No statistical methods were used to determine sample size, but the sample sizes were similar to those of previous reports[17]. No exclusion/inclusion criteria were applied to the mice used in this study. Group allocation and outcome assessment were performed in a blinded manner. In vitro experiments were repeated at least three times. All results are expressed as the mean ± standard error of the mean. Statistical significance was determined by using the two-tailed Student's $t$-test and $p < 0.05$ was considered to indicate a significant difference.

**Data availability.** The authors declare that all the data supporting the findings of this study are available within the article and its Supplementary Information Files. All MS raw data were stored in jPOSTrepo1 [https://repository.jpostdb.org/][50]. The project ID containing these data was JPST000398/PXD009147.

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

## Acknowledgments

We thank the members of Yamagata Laboratory for discussion and technical assistance. We thank Cheng Xu, Hiroki Ochi, Toru Fukuda, and Shingo Sato (Tokyo Medical and Dental University) for technical assistance with bone histomorphometric analyses. We also thank Osteopharma Inc. for supplying rhBMP-2. This study was (partially) supported by a Grant-in-Aid for Scientific Research (B) (16H05328) (K.Y.), (C) (17K11014) (T.Y.), and Scientific Research on Innovative Areas "Stem Cell Aging and Disease" (17H05650) from MEXT, Japan (T.Y.); by a grant from the Japan Agency for Medical Research and Development, AMED under Grant Number [JP17gm5010002] (K.Y.); by a grant from the Takeda Science Foundation (T.Y.); and by grants from the Japan Endocrine Society (T.Y.), the Naito Foundation (K.Y.), the Uehara Memorial Foundation (K.Y.), and the Astellas Foundation for Research on Metabolic Disorders (T.Y. and K.Y.).

## Author contributions

T.Y. and K.Y. conceived the project, designed the experiments, and wrote the manuscript. M.F., Md.F.K., S.U.S., W.K., D.K., H.O., M.T., K.O., T.S., Y.S., M.C., T.M., and Y.K. performed experiments and analyzed data. E.B. provided mice and discussed data. T.N., H.T., K.N., H.M., S.O., Y.Y., Y.A., Y.O., N.A., S.T., and H.M. provided critical supplies and support for experiments.

## Additional information

**Competing interests:** The authors declare no competing interests.

