## [Peer Review File · Nature Communications]

Reviewers' comments:

Reviewer #1 (Remarks to the Author):

This work addresses the role of Sirt7 in bone homeostasis in mice. Using murine models of global or osteoblast specific Sirt7 deletion the authors show that Sirt7 is required for physiological bone formation and, thereby, optimal bone mass. These findings are novel and significant. In addition, the authors provide data to suggest that Sirt7 increases osteoblast number by stimulating the activity of Osterix1, a critical transcription factor for osteoblastogenesis. Through extensive work in vitro, most of it in cell lines, the authors show that Sirt7 activates Osx1 by deacylation of lysine 368, which in turn facilitates the depropionylation of Osx1 by Sirt1. The data is, for its most part, robust and the paper is well written. However, there is no data to connect the in vivo findings with the in vitro mechanistic work. In addition, some of the features found in vivo indicate that the effect of Sirt7 might be due to effects other than the altered post-translational modifications of Osx1. The following is a list of concerns that should be addressed.

Major points:

1- There is no evidence, in primary cells from the Sirt7 deleted mice or in cell lines in which Sirt7 is silenced that endogenous Osx1 deacylation is altered. All mechanistic work of Fig. 5-7 is performed in cells overexpressing Osx1, Sirt7 or both. Therefore, it is unclear whether the Sirt7-induced post-translational modifications occur in the endogenous Osx1 or are the result of an artifact caused by the overexpression of the proteins. At least, the authors should examine whether Sirt7 and Osx1 interact in primary osteoblast and whether cells from Sirt7 deleted mice exhibit altered Osx1 lysine propionylation.

2- Sirt7 deletion caused a decrease in Runx2 expression (Fig. 2f). The reasons for this are unclear because Osx1 does not regulate Runx2. In fact, Baek et al (JBMR 2009, 24:1055) have shown that deletion of Osx1 in osteoblasts using Col1-Cre increases Runx2. Deletion of Osx1 also caused a decrease in osteoblast proliferation and only a mild decrease in mineralization of cells in vitro. In contrast Sirt7 deletion did not affect proliferation (Fig. 2b) and had a profound effect on osteoblast mineralization (Fig. 2a). These published evidence is not in line with the premise that deletion of Sirt7 decreases Osx1 activation. A decrease in Runx2 caused by Sirt7 deletion could explain the decrease in Osx1 expression and osteoblast number. At minimum, the authors should discuss the current findings in view of the known role of Osx1 in osteoblasts.

Other concerns:

1-The title should indicate the major findings of the work which is Sirt7 is important for bone formation.

2- It is unclear whether the WT and Sirt KO mice used in Fig. 1 are littermates.

Reviewer #2 (Remarks to the Author):

Lysine depropionylation of Osterix by sirtuins is important for bone formation Fukuda et al

In this manuscript Fukuda et al. have shown that one of the Sirtuins, Sirt7, plays an anabolic role on bone formation by regulating Osterix (Osx) transcriptional activity. The results indicate that Sirt7 KO female mice have less bone formation showing a mild bone phenotype. Cell culture experiments with loss of function by siRNA and Sirt7 KO primary osteoblasts demonstrate that Sirt7 promotes osteoblast differentiation and expression of osteoblast marker genes. To obtain insights in the mechanisms by which Sirt7 plays a positive regulatory role in osteoblast differentiation, several experiments are presented. Propionylation of several lysines residues of Osx attenuates its transcriptional activity. The model that is proposed is that interactions between Osx and Sirt7 would lead to deacylation of lysine 368 by Sirt7 and that this would facilitate the

depropionylation of lysine residues in Osx thereby enhancing its transcriptional activity. Although the identification of propionylation of Osx is an interesting observation its function in bone formation is not entirely clear.

Several major concerns need to be addressed to strengthen the biological significance of the depropionylation of Osx by Sirt7 in the regulation of osteoblast differentiation and bone formation. Sirtuins are generally anti-aging factors, their levels decrease during aging and several Sirtuin KO mutants have progeria like symptoms.

Another study (Vazquez et al. 2016) has indicated that Sirt7 KO mice show embryonic and perinatal lethality and age fast, raising the possibility that bone loss could be due to premature aging.

The paper does not provide evidence for acetylation of Osx by Sirt7 which weakens the proposed model for the mode of action of Sirt7 in controlling Osx activity.

Also it is noted that Sirt1 and Sirt7 both synergistically decrease propionylation of Osx, which raises the possibility for a role of Sirt1 in control of Osx activity. Due to functional redundancy among Sirtuins, the specificity of sirtuins in regulation of Osx should be clearly addressed.

The paper provides evidence and concludes that osteoclast differentiation is not affected in Sirt7 KO mice (Page 8, Line 187), In the abstract and on page 5 last paragraph a sentence reads "We found that Sirt7 KO mice developed severe osteopenia due to decreased bone formation along with an increase of osteoclasts." Again on page 7, line 151, it is indicated that SIRT7 also controls osteoclastogenesis (Suppl Fig 1k,l). These ambiguities need to be clarified.

The levels of Osx and the levels of expression of osteoblast marker genes should be measured in Sirt7 KO both during the early postnatal period and in adult mice to correlate the bone phenotype with the levels of osteoblast gene expression.

In figure 2b, osteoblast cell numbers were counted at 72h after seeding.. Sirtuins are known to induce apoptosis that trigger aging. Therefore, cell number analysis and levels of Sirt7 should be shown beyond 72 hr and at different time points to test whether Sirt7 levels progressively decrease and affect osteoblast proliferation and differentiation. The observation that Osx mRNA levels significantly decreased in Sirt7shRNA cells (Fig 2f) raises the possibility of another role of Sirt7 in regulation of Osx gene expression through chromatin remodeling. Sirt7 might regulate Osx activity at both transcriptional and post translational steps. Chromatin immunoprecipitation for the occupancies of Sirt7 at the Osx gene and H3K18Ac will provide needed information to assess the role of Sirt7 in remodeling of Osx chromatin and its effect on gene expression

Figure 3. Because there is no change in osteoclast differentiation in Sirt7 knock out mice, this figure should be placed in the supplementary section. In addition the text should be modified to remove the ambiguity whether Sirt7 KO leads to bone loss by decreasing Osx transcriptional activity and increasing osteoclast numbers. TRAP staining of skeletal tissues should be shown.

In mouse genetic studies detailed information is needed for the sake of clarity. In figure 4, Col1a1-Cre was used to inactivate the floxed alleles of Sirt7. This Cre driver becomes active during embryonic development when osteoblasts first differentiate (E15) and hence deletes the Sirt7 gene before birth.

Based on another report (Vazquez et al 2016) Sirt7 KO mice show perinatal lethality whereas the survivors also die prematurely. If this is the case in this study, the authors should discuss progeny survival . Complementary data are needed showing the levels of osteoblast specific gene expression to correlate with the levels of Sirt7.

A control is missing in Fig5 a and b. It is important to show the basal level of reporter activity without Osx expression. Figure 5c,d,e and f convincingly show interactions of Osx with Sirt7 and these experiments should have been done in the absence of NAM, but Suppl Figure 3b showed the interactions of Osx with Sirt7 only in the presence of NAM, but not in the absence of NAM Explanatory note provided by authors suggests that only acetylated Osx interacts with Sirt7. However the authors failed to provide evidence for acetylation of Osx either by mass spectrometry or western blot. They need to show experiments (similar to 5c and d) in the presence of NAM.

Figure 6a shows no acetylation of Osx, which is contradicting other results showing acetylation mediated interaction of Osx with Sirt7 (using NAM). It is also apparent that increased levels of Osx propionylation is due to higher levels of HA-Osx protein loaded in the blot. In figure 6e, decreased gene expression could also be due to toxicity of Na-Pro or a general inhibition of transcription by Na-Pro. Runx2 is an upstream regulator of Osx, then why was Runx2 expression decreased by Na-Pro if propionylation of Osx has an inhibitory effect on osteoblast gene expression. Authors should also test the effect of Na-Pro on the expression of nonosteoblast genes or perform a similar experiment in Osx shRNA cells. Fig 6g does provide a control for Osx, the basal level activity of the Col1a1- reporter is missing (a control without Osx). Prior to transfection of propionylated Osx it is important to show the extent of propionylated Osx either through western or mass spectrometry.

Reviewer #3 (Remarks to the Author):

The manuscript of Fukuda et al. demonstrates that the transcription factor Osterix (OSX) plays an important role in bone formation in mice. They show that SIRT7 interacts with OSX and it promotes its deacetylation, specifically propionylation. The authors claim that propionylation on OSX is preventing its full activity, and depropionylation by the complex SIRT1/SIRT7 is critical for osteoblast differentiation.

The work was well performed. The significance of the study is high (osteoporosis is a serious and widespread disease), and the results interestingly suggest that propionylation is a functionally important post-translational modification. This would be an important aspect as, although propionylation has been characterized on multiple proteins and organisms, its differential function from acetylation is not fully demonstrated.

My concern is that there is not much direct proof that only propionylation, and not acetylation, is responsible to block the function of OSX. The only direct evidence is Figure 6A, although it could be that the anti-acetyl antibody does not efficiently interact with acetylation on OSX. Nicotinamide (NAM) inhibits the enzymatic activity of sirtuins, so both acetylation and propionylation would be upregulated. The same can be said for the mutations, as both modifications cannot be catalyzed on the arginine residues. Figure 6E and 6F are not direct evidence either; sodium-propionate has a very similar structure to sodium-butyrate, which is a well known deacetylase inhibitor.

The identification of propionylation with mass spectrometry is a good evidence, but I see a few issues there: (i) the "accurate" analysis (using the Orbitrap Fusion) showed specific propionylation, but it was only achieved after treating cells for 24 hours with 50 mM sodium propionate. 50 mM is a lot, and it can be used to generate propionyl-CoA; the natural abundance of acetyl-CoA in the cell is around 10 uM (micromolar). I am wondering how much acetylation on OSX would have been detected after treating cells with 50 mM acetate. (ii) The authors did not provide enough details about the database searching; for instance, did they consider acetylation as dynamic modification? Also, raw files should be shared to allow the reviewer (and future readers) to use them. (iii) Propionylation was identified without sodium propionate treatment in the "non-accurate" analysis (the one with the ion trap). However, I am not sure how reliable these data can be; the ion trap requires very wide mass tolerances, which are usually not considered acceptable in the proteomics

community anymore, as it is known how much contamination is present even in a specific immunoprecipitation. Some spectra should be shown to prove the quality of these annotations.

Nevertheless, I think this work is important. I would not want to stall it for too long for this ambiguity acetylation/propionylation. I would recommend the authors to do at least one of the following, and then I would approve the work:

- Re-tune the manuscript (including the title) claiming that acylation, and not propionylation, cause reduced function of OSX;
- Show that acetylated OSX can be recognized by the antibody in use. It would be enough a synthetic peptide acetylated on the sites mentioned in the manuscript for propionylation;
- Show with mass spectrometry that growing cells with 50 mM acetate does not lead to OSX acetylation (and please share the raw files in public repositories);
- Provide evidence that no acetylation is detectable in the current analyses by mass spectrometry (including file sharing).

One more minor comment: I would specify somewhere that Osterix (OSX) is currently more known as SP7, and I would indicate somewhere the UniProt accession number where the sequence was retrieved: <http://www.uniprot.org/uniprot/Q8VI67>

Responses to the comments of Reviewer #1

We wish to thank the reviewer for the comment “The data is, for its most part, robust and the paper is well written” and for providing constructive suggestions.

We have addressed all the points raised by the reviewer through new experiments and/or new text. According to your suggestion (The title should indicate the major findings of the work which is *Sirt7* is important for bone formation) and that of reviewer #3 (Re-tune the manuscript (including the title) claiming that acylation, and not propionylation, cause reduced function of OSX), we have substantially rewritten the manuscript and the title to focus on control of bone formation by SIRT7 through regulation of the lysine acylation of Osterix (OSX). Therefore, results concerning the propionylation/acetylation of OSX have been moved to the last figure as additional parts.

“There is no evidence, in primary cells from the *Sirt7* deleted mice or in cell lines in which *Sirt7* is silenced that endogenous *Osx1* deacylation is altered. All mechanistic work of Fig. 5-7 is performed in cells overexpressing *Osx1*, *Sirt7* or both. Therefore, it is unclear whether the *Sirt7*-induced post-translational modifications occur in the endogenous *Osx1* or are the result of an artifact caused by the overexpression of the proteins. At least, the authors should examine whether *Sirt7* and *Osx1* interact in primary osteoblast and whether cells from *Sirt7* deleted mice exhibit altered *Osx1* lysine propionylation.”

We thank the reviewer for pointing this out. To address this judicious request, we have examined whether endogenous SIRT7 and OSX interact in primary osteoblasts and whether propionylation of endogenous OSX is enhanced in cells derived from *Sirt7* KO mice

As shown in Fig. 5d, interaction of endogenous SIRT7 and OSX in cultured primary osteoblasts was detected by the co-immunoprecipitation assay. We have added this information to page 11, paragraph 1. We also demonstrated that propionylation of endogenous OSX was enhanced in primary osteoblasts treated with NAM (Fig. 7a), in primary osteoblasts derived from *Sirt7* KO mice (*in vitro*) (Fig. 7b), and in the calvariae of *Sirt7* KO mice (*in vivo*) (Fig. 7c). Therefore, we have changed the text to reflect these

important new results (page 14, paragraph 1 and page 20, paragraph 1). Again, we thank the reviewer for this very important suggestion.

“Sirt7 deletion caused a decrease in Runx2 expression (Fig. 2f). The reasons for this are unclear because Osx1 does not regulate Runx2. In fact, Baek et al (JBMR 2009, 24:1055) have shown that deletion of Osx1 in osteoblasts using Coll-Cre increases Runx2. Deletion of Osx1 also caused a decrease in osteoblast proliferation and only a mild decrease in mineralization of cells in vitro. In contrast Sirt7 deletion did not affect proliferation (Fig. 2b) and had a profound effect on osteoblast mineralization (Fig. 2a). These published evidence is not in line with the premise that deletion of Sirt7 decreases Osx1 activation. A decrease in Runx2 caused by Sirt7 deletion could explain the decrease in Osx1 expression and osteoblast number. At minimum, the authors should discuss the current findings in view of the known role of Osx1 in osteoblasts.”

The reviewer is correct to state that some of our data did not seem to be in line with published results about OSX deficient mice. We have now added discussion about following two issues to page 18, paragraph 2.

A new possibility: We suggest that suppression of bone formation in *Sirt7* KO mice is mainly OSX dependent, because overexpression of mutant OSX (K368R) largely rescued impairment of mineralization in cultures of *Sirt7* KD MC3T3-E1 cells (Fig. 6e). However, we cannot exclude the possibility that SIRT7 regulates osteoblast differentiation by acting on non-OSX factors. Sirtuins have multiple substrates and regulate several signaling pathways in cells, so SIRT7 may act to increase *Runx2* expression at a certain stage of osteoblast differentiation.

Partial regulation of various OSX functions by SIRT7: OSX regulates the expression of target genes by several different mechanisms. We demonstrated that SIRT7 regulated the transcriptional activity of OSX mediated via GC-box DNA elements, but did not seem to affect its Dlx coactivator function (new data in Supplemental Fig. 3b and on page 10, paragraph 1). Thus, it can be suggested that SIRT7 only partially activates transactivation by OSX, so that OSX functionality is not completely abolished in *Sirt7* KO osteoblasts. Zhang *et al.* have suggested that OSX disrupts DNA binding by Tcf1, a partner of β -catenin, and that this is at least partly responsible for OSX-mediated

inhibition of osteoblast proliferation³⁶. Accordingly, SIRT7-dependent transactivation activity of OSX may not have an important role in osteoblast proliferation.

References

36. Zhang, C. *et al.* B. Inhibition of Wnt signaling by the osteoblast-specific transcription factor Osterix. *Proc. Natl. Acad. Sci. U. S. A.* **105**, 6936-6941 (2008).

“The title should indicate the major findings of the work which is Sirt7 in important for bone formation.”

We thank the reviewer for recognizing that our major finding is the importance of SIRT7 for bone formation. According to this suggestion and a suggestion from reviewer #3, we have now focused on control of bone formation by SIRT7 through regulation of the lysine acylation of Osterix. Therefore, we have changed the title to “SIRT7 has a critical role in bone formation by regulating lysine acylation of SP7/Osterix”.

“It is unclear whether the WT and Sirt KO mice used in Fig. 1 are littermates.”

Thank you for pointing this out. Breeding was only performed in the heterozygote state (*Sirt7*^{+/-} × *Sirt7*^{+/-}) and littermates (WT and *Sirt7* KO mice) were used for the studies. We have now added this information to the manuscript on page 21, paragraph 2.

Responses to the comments of Reviewer #2

We wish to thank the reviewer for many helpful comments and constructive suggestions.

We have addressed all the points raised by the reviewer through new experiments and/or by adding new text. According to the suggestions of reviewer #1 (The title should indicate the major findings of the work which is *Sirt7* in important for bone formation) and reviewer #3 (Re-tune the manuscript (including the title) claiming that acylation, and not propionylation, cause reduced function of OSX), we have substantially rewritten the manuscript and the title to focus on control of bone formation by SIRT7 through regulation of the lysine acylation of Osterix. Data concerning the propionylation/acetylation of Osterix have been moved to the last figure as additional parts.

“Sirtuins are generally anti-aging factors, their levels decrease during aging and several Sirtuin KO mutants have progeria like symptoms. Another study (Vazquez et al. 2016) has indicated that Sirt7 KO mice show embryonic and perinatal lethality and age fast, raising the possibility that bone loss could be due to premature aging.”

So far, three independent *Sirt7* KO mouse lines have been reported. Vazquez obtained *Sirt7* KO mice line of Dr. Danica Chen’s group (University of California, Berkeley, USA) and 129sv *Sirt7* KO mice were backcrossed with C57BL/6 using a speed congenic strategy. However, Chen’s group did not particularly mention premature aging in their first report (Shin J. *et al. Cell Rep.* **5**, 654-665 (2013)). Dr. Johan Auwerx (Ecole Polytechnique Fédérale de Lausanne, Switzerland) reported that the offspring of heterozygous *Sirt7* breeders (*Sirt7*^{+/-}) showed a normal Mendelian ratio (+/+:+/-:-/- = 22.8%:52.4%:24.9%) and normal sex ratio (male:female = 48%:52%), and that body weight and body composition were indistinguishable between both genotypes (Ryu D. *et al. Cell Metab.* **20**, 1-14 (2014)).

In our *Sirt7* KO mice, breeding heterozygous *Sirt7* mice resulted in a normal Mendelian ratio and sex ratio (Fig. 1j), and there were no significant differences of postnatal lethality or body weight between *Sirt7* KO mice and WT mice (Fig. 1k,l). Therefore, no apparent embryonic lethality, postnatal lethality, and growth retardation

has been found in at least two independent *Sirt7* KO mouse lines, including our line. These data indicate that the low bone mass phenotype of *Sirt7* KO mice is not due to premature aging. We have now added this information to page 6, paragraph 1. We have no explanation as to why only one line of *Sirt7* KO mice shows a progeroid phenotype.

“The paper does not provide evidence for acetylation of Osx by Sirt7 which weakens the proposed model for the mode of action of Sirt7 in controlling Osx activity.”

The reviewer is right. We could not detect acetylation of OSX by western blotting before, so we did not come to a conclusion as to whether SIRT7 deacetylated OSX or not.

To test whether the antibody could detect acetylated OSX in osteoblasts, we first confirmed that acetylation of histone H3 (by treatment with NAM) and OSX (by cotransfection with p300) were clearly detected by our western blotting system (Supplementary Fig. 5a,b). While these western blotting conditions (exposure time: several seconds) could not detect acetylation of OSX in osteoblasts, when we used a much longer exposure time (several minutes), we could just detect acetylation of OSX in primary osteoblasts (Fig. 7a) and also in MC3T3-E1 cells with stable overexpression of HA-OSX (Supplementary Fig. 5c). Now we have shown that acetylated Osterix is detected by western blotting and mass spectrometry (Fig. 7a-c, Supplemental Fig. 5c, and text on page 14, paragraph 1; and Supplemental Figure 6 and text on page 15, paragraph 1), and that acetylation of Osterix is not altered by SIRT7 deficiency (Fig. 7b,c, and text on page 14, paragraph 1).

We demonstrated that SIRT7 activates OSX transactivation activity through deacylation of lysine 368, but we could not identify the actual modification of lysine 368 by MS analysis. Recent studies have shown that SIRT7 has deacetylase activity, desuccinylase activity, and defatty-acylase activity (removing myristoylation)^{16,33}. Acetylation of OSX was not changed in the calvariae of *Sirt7* KO mice (Fig. 7c), suggesting it is unlikely that SIRT7 deacetylates lysine 368. Further investigation will be necessary to clarify the actual modification of lysine 368 for better understanding of the molecular mechanism through which SIRT7 regulates the transactivation activity of OSX. We have added this information to page 17, paragraph 2.

References

16. Li, L. *et al.* SIRT7 is a histone desuccinylase that functionally links to chromatin compaction and genome stability. *Nat. Commun.* **7**, 12235 (2016).
33. Tong, Z. *et al.* SIRT7 Is an RNA-Activated Protein Lysine Deacylase. *ACS Chem. Biol.* **12**, 300-310 (2017).

“Also it is noted that Sirt1 and Sirt7 both synergistically decrease propionylation of Osx, which raises the possibility for a role of Sirt1 in control of Osx activity. Due to functional redundancy among Sirtuins, the specificity of sirtuins in regulation of Osx should be clearly addressed.”

Thank you for pointing this out. We have already shown that SIRT1 and SIRT7 synergistically promote the transcriptional activity of OSX (Fig. 7k). We have now mentioned the role of SIRT1 in regulating OSX activity, in addition to other molecular mechanisms by which SIRT1 could regulate osteoblast differentiation (page 17, paragraph 3) as follows: “In the present study, we demonstrated synergistic depropionylation of OSX by SIRT1 and SIRT7 (Fig. 7j), resulting in activation of OSX transactivation (Fig. 7k). SIRT1 has already been reported to modulate bone formation by osteoblasts. In MSC-specific *Sirt1* KO mice, it was reported that SIRT1 regulates osteoblastic differentiation of MSCs by deacetylation of β -catenin¹². Depletion of SIRT1 in osteoblast progenitors employing *Osx-Cre* mice led to a decrease in cortical bone thickness associated with decreased bone formation, resulting from increased sequestration of β -catenin by acetylated-FoxOs¹⁰. In addition to these mechanisms, our present findings suggest that SIRT1 positively regulates osteoblast differentiation by modulating the propionylation of OSX.”

Moreover, to clearly address the specific roles of SIRT1 and SIRT7 in regulation of OSX, we have proposed a model for regulation of OSX transactivation activity by SIRT7 and SIRT1 through lysine deacylation (Supplementary Fig. 7b).

References

10. Iyer, S. *et al.* Sirtuin1 (Sirt1) promotes cortical bone formation by preventing β -catenin sequestration by FoxO transcription factors in osteoblast progenitors. *J. Biol. Chem.* **289**, 24069-24078 (2014).

12. Simic, P. *et al.* SIRT1 regulates differentiation of mesenchymal stem cells by deacetylating β -catenin. *EMBO Mol. Med.* **5**, 430-440 (2013).

“The paper provides evidence and concludes that osteoclast differentiation is not affected in Sirt7 KO mice (Page 8, Line 187). In the abstract and on page 5 last paragraph a sentence reads “We found that Sirt7 KO mice developed severe osteopenia due to decreased bone formation along with an increase of osteoclasts.” Again on page 7, line 151, it is indicated that SIRT7 also controls osteoclastogenesis (Suppl Fig 1k,l). These ambiguities need to be clarified.”

We apologize for our confusing explanation in the original manuscript. The experiment described on page 8, line 187 of the original manuscript is an *in vitro* osteoblast-free osteoclast differentiation assay, and is not *in vivo* data obtained from mice. To avoid misunderstanding, we have changed the sentence “there were no significant differences between *Sirt7* KO and WT mice” to “there were no significant differences between osteoclasts derived from monocytes/macrophages of *Sirt7* KO and WT mice” (page 8, paragraph 2).

We found that the number of osteoclasts was increased in the lumbar spine of *Sirt7* KO mice compared with WT controls (*in vivo*) (Supplementary Fig. 1n,o,p), but SIRT7 was not essential for osteoclastogenesis in cell culture (*in vitro*) (Supplementary Fig. 2). We have already discussed the reason for this discrepancy in the original manuscript on page 16, paragraph 3 (page 19, paragraph 2 of the revised manuscript).

“The levels of *Osx* and the levels of expression of osteoblast marker genes should be measured in Sirt7 KO both during the early postnatal period and in adult mice to correlate the bone phenotype with the levels of osteoblast gene expression.”

To address this issue, we analyzed the expression of osteoblastic marker genes in the femurs of *Sirt7* KO mice at 14-15 weeks old (adulthood) and at 15 days old (early postnatal period). In adult *Sirt7* KO mice, consistent with the severe phenotype, expression of *alkaline phosphatase, liver/bone/kidney (Alp)*, *collagen, type I, alpha 1 (Colla1)*, *osteocalcin (Ocn)*, *Osx*, and *Runx2* were all significantly reduced (Fig. 2e and text on page 7, paragraph 1). In the early postnatal period, expression of some

osteoblastic marker genes tended to be lower in *Sirt7* KO mice (Supplementary Fig. 1m and text on page 7, paragraph 1). Consistent with the mild attenuation of gene expression, μ CT analysis showed that 15-day old *Sirt7* KO mice had less severe bone changes than adult mice (Supplementary Fig. 1i-l and text on page 6, paragraph 1).

“In figure 2b, osteoblast cell numbers were counted at 72h after seeding. Sirtuins are known to induce apoptosis that trigger aging. Therefore, cell number analysis and levels of Sirt7 should be shown beyond 72 hr and at different time points to test whether Sirt7 levels progressively decrease and affect osteoblast proliferation and differentiation.”

The original Figure 2b showed a proliferation assay using primary calvarial osteoblasts from *Sirt7* KO and WT mice. The level of SIRT7 in osteoblasts from *Sirt7* KO mice is always ZERO. Therefore, we cannot comment on the relationship between a progressive decrease of the *Sirt7* level and osteoblast proliferation/differentiation from the results of these experiments. Just in case, we have added the cell numbers on day 5, which is the over-confluent stage (Fig. 3b). Cell numbers were not decreased compared with day 3 and there were no differences between cells from *Sirt7* KO and WT mice.

Vazquez *et al.* reported “we did not observe major differences in the cell growth and cell cycle profiles between *Sirt7*^{-/-} and WT MEFs, in agreement with previous reports (Vakhrusheva *et al.*, 2008a)”. Fang *et al.* reported “we did not observe any overt changes in the proliferation capacity of *Sirt7*-deficient preadipocytes”. Therefore, the growth of primary cultured *Sirt7* KO cells was not necessarily altered.

References

- Vazquez, BN. *et al.* SIRT7 promotes genome integrity and modulates non-homologous end joining DNA repair. *EMBO J.* **35**, 1488-1503 (2016).
- Vakhrusheva, O., Braeuer, D., Liu, Z., Braun, T., Bober, E. Sirt7-dependent inhibition of cell growth and proliferation might be instrumental to mediate tissue integrity during aging. *J. Physiol. Pharmacol.* **59**(Suppl 9), 201–212 (2008a).
- Fang, J. *et al.* Sirt7 promotes adipogenesis in the mouse by inhibiting autocatalytic activation of Sirt1. *Proc. Natl. Acad. Sci. USA.* **114**, E8352-E8361 (2017).

“The observation that *Osx* mRNA levels significantly decreased in *Sirt7*shRNA cells (Fig 2f) raises the possibility of another role of *Sirt7* in regulation of *Osx* gene expression through chromatin remodeling. *Sirt7* might regulate *Osx* activity at both transcriptional and post translational steps. Chromatin immunoprecipitation for the occupancies of *Sirt7* at the *Osx* gene and H3K18Ac will provide needed information to assess the role of *Sirt7* in remodeling of *Osx* chromatin and its effect on gene expression.”

We thank the reviewer for pointing this out. We now suggest that suppression of bone formation in *Sirt7* KO mice is mainly OSX dependent, because overexpression of mutant OSX (K368R) largely reversed impairment of mineralization in cultures of *Sirt7* KD MC3T3-E1 cells (Fig. 6e). However, we cannot exclude the possibility that SIRT7 regulates osteoblast differentiation by acting on non-OSX factors. Sirtuins have multiple substrates and regulate several signaling pathways in cells, so SIRT7 may activate the transcription factor regulating the *Osx* gene at a certain stage of osteoblast differentiation. We have added discussion about this possible effect of SIRT7 in osteoblasts to page 18, paragraph 2.

We agree that chromatin immunoprecipitation experiments to assess occupancy of the *Osx* gene by SIRT7 may potentially provide additional information about other functions of SIRT7. However, we have already provided much data showing that SIRT7 regulates Osterix transcriptional activity, and ChIP-Seq analysis of SIRT7 would be a major undertaking that lies outside the scope of this study.

“Figure 3. Because there is no change in osteoclast differentiation in *Sirt7* knock out mice, this figure should be placed in the supplementary section. In addition the text should be modified to remove the ambiguity whether *Sirt7* KO leads to bone loss by decreasing *Osx* transcriptional activity and increasing osteoclast numbers. TRAP staining of skeletal tissues should be shown.”

As mentioned in our answer to another query you made, we found that the number of osteoclasts was increased in the lumbar spine of *Sirt7* KO mice compared with WT controls (*in vivo*) (Supplementary Fig. 1n,o,p). Therefore, we still consider that SIRT7 is essential for osteoblast differentiation and osteoclast formation *in vivo*. On

the other hand, SIRT7 is not essential for osteoclastogenesis *in vitro* (Supplementary Fig. 2). We have discussed the possible mechanism by which SIRT7 regulates osteoclast formation *in vivo* on page 19, paragraph 2.

To focus on the role of SIRT7 in bone formation, Figure 3 in the original manuscript has been moved to Supplemental Figure 2. Representative TRAP staining of skeletal tissue for bone histomorphometric analysis (Supplementary Fig. 1n,o), is shown in supplemental Figure 1p.

“In mouse genetic studies detailed information is needed for the sake of clarity. In figure 4, Colla1-Cre was used to inactivate the floxed alleles of Sirt7. This Cre driver becomes active during embryonic development when osteoblasts first differentiate (E15) and hence deletes the Sirt7 gene before birth.”

Thank you for pointing this out. We have added the information about *Colla1-Cre* mice (page 21, paragraph 2).

“Based on another report (Vazquez et al 2016) Sirt7 KO mice show perinatal lethality whereas the survivors also die prematurely. If this is the case in this study, the authors should discuss progeny survival. Complementary data are needed showing the levels of osteoblast specific gene expression to correlate with the levels of Sirt7.”

As mentioned above, there was no apparent embryonic lethality, postnatal lethality, or growth retardation in two independent *Sirt7* KO mouse lines, including our line.

To show that the level of osteoblast-specific gene expression was correlated with the *Sirt7* level, we analyzed the expression of osteoblastic marker genes in *Sirt7* KO mice and aged WT mice. As mentioned in our reply to another query of yours, expression of osteoblastic marker genes was significantly reduced in *Sirt7* KO mice (Fig. 2e). Expression of *Sirt6* and *Sirt7* was reduced in the bones of aged mice (Fig. 2f), while expressions of osteoblastic marker genes was markedly decreased (Supplementary Fig. 1q). Thus, attenuation of these sirtuins may contribute to the weakness of aged bone. We have now added this information to page 7, paragraph 2.

“A control is missing in Fig5 a and b. It is important to show the basal level of reporter activity without Osx expression.”

We apologize for oversimplification of our presentation in the initial manuscript. To address this concern, we have added all the controls missing from the original Fig. 5a, b (revised Fig. 5a,b).

“Figure 5c,d,e and f convincingly show interactions of Osx with Sirt7 and these experiments should have been done in the absence of NAM, but Suppl Figure 3b showed the interactions of Osx with Sirt7 only in the presence of NAM, but not in the absence of NAM. Explanatory note provided by authors suggests that only acetylated Osx interacts with Sirt7. However the authors failed to provide evidence for acetylation of Osx either by mass spectrometry or western blot. They need to show experiments (similar to 5c and d) in the presence of NAM.”

We apologize for our inaccurate explanation in the original manuscript. First of all, sirtuins interact with acylated substrates, not only with acetylated substrates, and NAM inhibits sirtuin deacylases activity, not only deacetylase activity. We have more carefully explained the principles of our experiments and why we used NAM to show that SIRT7 and OSX are in an enzyme-substrate relation (page 11, paragraph 2) as follows:

“To show that OSX is a substrate for SIRT7 as an enzyme, we performed the pull-down assay with acylated or underacylated OSX. Since binding affinity between an enzyme and its substrate generally decreases after the enzymatic reaction has finished, we considered that acylated OSX would show stronger binding to SIRT7 than deacylated OSX. As expected, endogenous OSX derived from MC3T3-E1 cells treated with nicotinamide (NAM), which inhibits the deacylase activity of sirtuins, was bound to Halo-SIRT7 beads, but endogenous OSX derived from MC3T3-E1 cells without NAM treatment showed little binding (Supplementary Fig. 3c). These results suggested that an enzyme-substrate relationship exists between SIRT7 and OSX. We also found an interaction between Halo-SIRT7 beads and OSX in HEK293T cells without NAM

treatment, presumably due to insufficient deacylation of OSX by its overexpression (Supplementary Fig. 3c).”

“Figure 6a shows no acetylation of Osx, which is contradicting other results showing acetylation mediated interaction of Osx with Sirt7 (using NAM).”

As mentioned above, Sirtuins interact with acylated substrates and NAM is inhibits sirtuin deacylases activity, not only deacetylase activity. Therefore, these results are not contradictory. To avoid confusion between “acylation” and “acetylation”, we have carefully explained acyl-lysine modifications in the Introduction (page 4, paragraph 2) as follows: “Sirtuins (SIRT1-7 in mammals) are nicotinamide adenine dinucleotide (NAD⁺)-dependent lysine deacylases that regulate a wide variety of biological process^{7,8}. Although sirtuins were thought to only act as lysine deacetylases, recent studies have revealed that these enzymes can also remove other acyl-lysine modifications, including propionylation, succinylation, malonylation, myristoylation, and palmitoylation.”

References

7. Guarente, L. Sirtuins, aging, and medicine. *N. Engl. J. Med.* **364**, 2235–2244 (2011).
8. Houtkooper, R.H., Pirinen, E., and Auwerx, J. Sirtuins as regulators of metabolism and healthspan. *Nature Reviews, Molecular Cell Biology* **13**, 225–238 (2012).

“It is also apparent that increased levels of Osx propionylation is due to higher levels of HA-Osx protein loaded in the blot.”

The reviewer is right to mention that precipitated HA-OSX protein was slightly high in the original Figure 6a. To address this point, we performed three independent experiments and statistically compared the propionylated OSX/total OSX ratio (see Figure 1 for reviewer #2). The data indicated that propionylation of OSX was strongly enhanced by NAM treatment. Furthermore, we have now provided more convincing evidence using primary osteoblasts (Fig. 7a), and the original Figure 6a has been moved to supplemental Figure 5c.

Figure 1 for reviewer #2.

“In figure 6e, decreased gene expression could also be due to toxicity of Na-Pro or a general inhibition of transcription by Na-Pro. Runx2 is an upstream regulator of Osx, then why was Runx2 expression decreased by Na-Pro if propionylation of Osx has an inhibitory effect on osteoblast gene expression. Authors should also test the effect of Na-Pro on the expression of nonosteoblast genes or perform a similar experiment in Osx shRNA cells.”

The reviewer is absolutely right. Na-prop treatment not only affects OSX, but may also affect many other factors, including the transcription factor regulating the *Runx2* gene. Reviewer #3 also pointed out that Figure 6e is not direct evidence. Therefore, we have moved the original Figure 6e to Supplemental Figure 5f. Furthermore, we have shown that at least the expression of non-osteoblast genes [*beta-2 microglobulin (B2m)*, *actin, beta (Actb)*, and *18S ribosomal RNA (18S)*] was unchanged.

“Fig 6g does provide a control for Osx, the basal level activity of the Col1a1- reporter is missing (a control without Osx). Prior to transfection of propionylated Osx it is important to show the extent of propionylated Osx either through western or mass spectrometry.”

We apologize for the lack of a control without OSX in the initial manuscript.

To address this concern, we performed this experiment again with a control (Fig. 7h). Transfected proteins were confirmed by western blotting using anti-OSX and anti-propionyllysine antibodies (Fig. 7h (right) and text on page 43, paragraph 2).

Responses to the comments of Reviewer #3

We wish to thank the reviewer for the encouraging comments [“The work was well performed. The significance of the study is high.” “I think this work is important.”] and for providing constructive suggestions.

We have now addressed all the points raised by the reviewer through new experiments and/or by adding text. According to the suggestions of reviewer #1 (The title should indicate the major findings of the work which is *Sirt7* in important for bone formation) and your suggestion (Re-tune the manuscript (including the title) claiming that acylation, and not propionylation, cause reduced function of OSX), we have substantially rewritten the manuscript and the title to focus on control of bone formation by SIRT7 through regulation of the lysine acylation of Osterix. Data concerning the propionylation/acetylation of Osterix have been moved to the last figure as additional parts.

“My concern is that there is not much direct proof that only propionylation, and not acetylation, is responsible to block the function of OSX. The only direct evidence is Figure 6A, although it could be that the anti-acetyl antibody does not efficiently interact with acetylation on OSX. Nicotinamide (NAM) inhibits the enzymatic activity of sirtuins, so both acetylation and propionylation would be upregulated. The same can be said for the mutations, as both modifications cannot be catalyzed on the arginine residues. Figure 6E and 6F are not direct evidence either; sodium-propionate has a very similar structure to sodium-butyrate, which is a well known deacetylase inhibitor.”

The reviewer is absolutely right. We showed that OSX undergoes lysine propionylation and propionylation reduced the transactivation activity of OSX (Fig. 7). However, we could not uncover differences of transactivation between propionylation and acetylation of OSX, because we were unable to prepare acetylated (but not propionylated) recombinant OSX for the luciferase assay. Due to this weakness, we have followed the reviewer’s helpful suggestion to “re-tune the manuscript claiming that acylation, not propionylation, cause reduced function of OSX”.

In contrast of functional difference, we have newly demonstrated that these two closely related acylation marks are differentially regulated by SIRT7. Propionylation of endogenous OSX, but not acetylation (we barely detected acetylation of OSX, see next paragraph), was increased in primary osteoblasts and calvariae obtained from *Sirt7* KO mice (new data, Fig. 7b,c). Further investigations will be needed to define the molecular mechanisms involved. It is possible that SIRT7-facilitated deacetylation of lysine residues in OSX affects a limited number of several acetylated lysine residues, so that small changes due to SIRT7 are masked in whole. We have added this comment to the Discussion on page 20, paragraph 1.

We tested whether the antibody could recognize acetylated OSX. When HEK293T cells were transfected with the 3×HA-OSX expression plasmid and p300-myc expression plasmid, acetylated OSX was strongly detected by our western blotting system (Supplementary Fig. 5b). These western blotting conditions (exposure time: several seconds) could not detect acetylation of OSX in osteoblasts, as we found before. However, using a much longer exposure time (several minutes), we could detect low levels of OSX acetylation in primary osteoblasts (Fig. 7a) and also in MC3T3-E1 cells with stable overexpression of HA-OSX (Supplementary Fig. 5c). We could also demonstrate that acetylation of OSX was slightly increased by treatment with NAM (Fig. 7a, Supplementary Fig. 5c, and text on page 14, paragraph 1).

Concerning the Na-prop treatment, the reviewer is absolutely right that it not only affects OSX, but also affects other factors. Reviewer #2 also pointed out that Figure 6e is not direct evidence. Therefore, we have moved the original Figure 6e to Supplemental Figure 5f. In addition, we have shown that at least the expression of non-osteoblast related genes [*beta-2 microglobulin (B2m)*, *actin, beta (Actb)*, and *18S ribosomal RNA (18S)*] was unchanged.

“The identification of propionylation with mass spectrometry is a good evidence, but I see a few issues there: (i) the “accurate” analysis (using the Orbitrap Fusion) showed specific propionylation, but it was only achieved after treating cells for 24 hours with 50 mM sodium propionate. 50 mM is a lot, and it can be used to generate propionyl-CoA; the natural abundance of acetyl-CoA in the cell is around 10 uM (micromolar). I am wondering how much acetylation on OSX would have been

detected after treating cells with 50 mM acetate. (ii) The authors did not provide enough details about the database searching; for instance, did they consider acetylation as dynamic modification? Also, raw files should be shared to allow the reviewer (and future readers) to use them. (iii) Propionylation was identified without sodium propionate treatment in the “non-accurate” analysis (the one with the ion trap). However, I am not sure how reliable these data can be; the ion trap requires very wide mass tolerances, which are usually not considered acceptable in the proteomics community anymore, as it is known how much contamination is present even in a specific immunoprecipitation. Some spectra should be shown to prove the quality of these annotations.”

The reviewer is right to state that 50 mM sodium propionate, which was used to generate propionyl-CoA, is a high concentration. As the reviewer knows well, it is hard to detect natural levels of acyl-modification by mass spectrometry. Therefore, we used an unphysiological concentration of sodium propionate to clearly detect the sites of propionylation on OSX. We tried to detect propionylation of OSX in cells without sodium propionate treatment by using Orbitrap Fusion mass spectrometry, but we did not succeed due to technical limitations. Concerning the “non-accurate” analysis, we discussed this with several mass spectrometry experts and decided to delete the results obtained by “non-accurate” analysis.

We apologize for our explanation in the original manuscript not being detailed enough. Since we could not detect acetylation of OSX by western blotting before, we simply did not focus on acetylation. We have identified several acetylated lysines in OSX (Supplemental Figure 6a), and all MS raw data were stored in jPOSTrepo1 (<https://repository.jpostdb.org/>). The project ID containing these data was JPST000398/PXD009147. We have no adequate explanation as to why western blotting for acetylated OSX was less efficient than mass spectrometry.

“Nevertheless, I think this work is important. I would not want to stall it for too long for this ambiguity acetylation/propionylation. I would recommend the authors to do at least one of the following, and then I would approve the work:

- Re-tune the manuscript (including the title) claiming that acylation, and not propionylation, cause reduced function of OSX;

- *Show that acetylated OSX can be recognized by the antibody in use. It would be enough a synthetic peptide acetylated on the sites mentioned in the manuscript for propionylation;*
- *Show with mass spectrometry that growing cells with 50 mM acetate does not lead to OSX acetylation (and please share the raw files in public repositories);*
- *Provide evidence that no acetylation is detectable in the current analyses by mass spectrometry (including file sharing)."*

We greatly appreciated these recommendations. We have addressed all four points, especially the first point as mentioned above. We have substantially rewritten the manuscript (including the title) to state that acylation, not propionylation, cause reduced function of OSX. For the second point, we showed that acetylated OSX was strongly detected by our western blotting system with the current antibody (Supplementary Fig. 5b). Concerning points 3 and 4, we have identified several acetylated lysines on OSX by mass spectrometry (Supplemental Figure 6), and all MS raw data were stored in jPOSTrepo1 (<https://repository.jpostdb.org/>). The project ID containing these data was JPST000398/PXD009147. We believe that the manuscript has been greatly improved by these modifications and we sincerely hope that you approve of our work.

“One more minor comment: I would specify somewhere that Osterix (OSX) is currently more known as SP7, and I would indicate somewhere the UniProt accession number where the sequence was retrieved: <http://www.uniprot.org/uniprot/Q8VI67>”

Thank you for pointing this out. We have specified both SP7 and Osterix in the title, abstract, and introduction. The UniProt accession number has been indicated in the Methods section (page 22, paragraph 1).

REVIEWERS' COMMENTS:

Reviewer #1 (Remarks to the Author):

The authors addressed adequately the concerns raised and substantially improved the manuscript. I have no further comments.

Reviewer #3 (Remarks to the Author):

The authors have made a remarkable effort in this manuscript, and they have answered to all the criticisms raised by this reviewer.

I have no more comments, and thus I recommend publication.

Reviewer #4 (Remarks to the Author):

The authors have addressed the reviewer's concerns appropriately.

I recommend publication of the manuscript.

Responses to the comments of Reviewer #1

REVIEWERS' COMMENTS:

The authors addressed adequately the concerns raised and substantially improved the manuscript. I have no further comments.

We wish to thank the reviewer for the acceptance. Important new results according to your judicious requests greatly improved our manuscript. Again, we appreciate the reviewer for fair peer review.

Responses to the comments of Reviewer #3

REVIEWERS' COMMENTS:

The authors have made a remarkable effort in this manuscript, and they have answered to all the criticisms raised by this reviewer. I have no more comments, and thus I recommend publication.

We wish to thank the reviewer for the acceptance. Your suggestions substantially changed our study and improved our manuscript. Again, we greatly appreciate the reviewer for fair peer review.

Responses to the comments of Reviewer #4

REVIEWERS' COMMENTS:

The authors have addressed the reviewer's concerns appropriately. I recommend publication of the manuscript.

We wish to thank the reviewer for the fair peer review and acceptance.